# DEEP REWIRING: TRAINING VERY SPARSE DEEP NETWORKS

**Guillaume Bellec, David Kappel, Wolfgang Maass & Robert Legenstein**
Institute for Theoretical Computer Science
Graz University of Technology
Austria
`{bellec,kappel,maass,legenstein}@igi.tugraz.at`

## ABSTRACT

Neuromorphic hardware tends to pose limits on the connectivity of deep networks that one can run on them. But also generic hardware and software implementations of deep learning run more efficiently for sparse networks. Several methods exist for pruning connections of a neural network after it was trained without connectivity constraints. We present an algorithm, DEEP R, that enables us to train directly a sparsely connected neural network. DEEP R automatically rewires the network during supervised training so that connections are there where they are most needed for the task, while its total number is all the time strictly bounded. We demonstrate that DEEP R can be used to train very sparse feedforward and recurrent neural networks on standard benchmark tasks with just a minor loss in performance. DEEP R is based on a rigorous theoretical foundation that views rewiring as stochastic sampling of network configurations from a posterior.

## 1 INTRODUCTION

Network connectivity is one of the main determinants for whether a neural network can be efficiently implemented in hardware or simulated in software. For example, it is mentioned in Jouppi et al. (2017) that in Google's tensor processing units (TPUs), weights do not normally fit in on-chip memory for neural network applications despite the small 8 bit weight precision on TPUs. Memory is also the bottleneck in terms of energy consumption in TPUs and FPGAs (Han et al., 2017; Iandola et al., 2016). For example, for an implementation of a long short term memory network (LSTM), memory reference consumes more than two orders of magnitude more energy than ALU operations (Han et al., 2017). The situation is even more critical in neuromorphic hardware, where either hard upper bounds on network connectivity are unavoidable (Schemmel et al., 2010; Merolla et al., 2014) or fast on-chip memory of local processing cores is severely limited, for example the 96 MByte local memory of cores in the SpiNNaker system (Furber et al., 2014). This implementation bottleneck will become even more severe in future applications of deep learning when the number of neurons in layers will increase, causing a quadratic growth in the number of connections between them.

Evolution has apparently faced a similar problem when evolving large neuronal systems such as the human brain, given that the brain volume is dominated by white matter, i.e., by connections between neurons. The solution found by evolution is convincing. Synaptic connectivity in the brain is highly dynamic in the sense that new synapses are constantly rewired, especially during learning (Holtmaat et al., 2005; Stettler et al., 2006; Attardo et al., 2015; Chambers & Rumpel, 2017). In other words, rewiring is an integral part of the learning algorithms in the brain, rather than a separate process.

We are not aware of previous methods for simultaneous training and rewiring in artificial neural networks, so that they are able to stay within a strict bound on the total number of connections throughout the learning process. There are however several heuristic methods for pruning a larger network (Han et al., 2015b;a; Collins & Kohli, 2014; Yang et al., 2015; Srinivas & Babu, 2015), that is, the network is first trained to convergence, and network connections and / or neurons are pruned only subsequently. These methods are useful for downloading a trained network on neuromorphic hardware, but not for on-chip training. A number of methods have been proposed that are capable of reducing connectivity during training (Collins & Kohli, 2014; Jin et al., 2016; Narang et al.,

2017). However, these algorithms usually start out with full connectivity. Hence, besides reducing computational demands only partially, they cannot be applied when computational resources (such as memory) is bounded throughout training.

Inspired by experimental findings on rewiring in the brain, we propose in this article deep rewiring (DEEP R), an algorithm that makes it possible to train deep neural networks under strict connectivity constraints. In contrast to many previous pruning approaches that were based on heuristic arguments, DEEP R is embedded in a thorough theoretical framework. DEEP R is conceptually different from standard gradient descent algorithms in two respects. First, each connection has a predefined sign. Specifically, we assign to each connection $k$ a connection parameter $\theta_k$ and a constant sign $s_k \in \{-1, 1\}$. For non-negative $\theta_k$, the corresponding network weight is given by $w_k = s_k \theta_k$. In standard backprop, when the absolute value of a weight is moved through $0$, it becomes a weight with the opposite sign. In contrast, in DEEP R a connection vanishes in this case ($w_k = 0$), and a randomly drawn other connection is tried out by the algorithm. Second, in DEEP R, gradient descent is combined with a random walk in parameter space (de Freitas et al., 2000; Welling & Teh, 2011). This modification leads to important functional differences. In fact, our theoretical analysis shows that DEEP R jointly samples network weights and the network architecture (i.e., network connectivity) from the posterior distribution, that is, the distribution that combines the data likelihood and a specific connectivity prior in a Bayes optimal manner. As a result, the algorithm continues to rewire connections even when the performance has converged. We show that this feature enables DEEP R to adapt the network connectivity structure online when the task demands are drifting.

We show on several benchmark tasks that with DEEP R, the connectivity of several deep architectures — fully connected deep networks, convolutional nets, and recurrent networks (LSTMs) — can be constrained to be extremely sparse throughout training with a marginal drop in performance. In one example, a standard feed forward network trained on the MNIST dataset, we achieved good performance with 2 % of the connectivity of the fully connected counterpart. We show that DEEP R reaches a similar performance level as state-of-the-art pruning algorithms where training starts with the full connectivity matrix. If the target connectivity is very sparse (a few percent of the full connectivity), DEEP R outperformed these pruning algorithms.

## 2 REWIRING IN DEEP NEURAL NETWORKS

Stochastic gradient descent (SGD) and its modern variants (Kingma & Ba, 2014; Tieleman & Hinton, 2012) implemented through the Error Backpropagation algorithm is the dominant learning paradigm of contemporary deep learning applications. For a given list of network inputs $\mathbf{X}$ and target network outputs $\mathbf{Y}^*$, gradient descent iteratively moves the parameter vector $\boldsymbol{\theta}$ in the direction of the negative gradient of an error function $E_{\mathbf{X}, \mathbf{Y}^*}(\boldsymbol{\theta})$ such that a local minimum of $E_{\mathbf{X}, \mathbf{Y}^*}(\boldsymbol{\theta})$ is eventually reached.

A more general view on neural network training is provided by a probabilistic interpretation of the learning problem (Bishop, 2006; Neal, 1992). In this probabilistic learning framework, the deterministic network output is interpreted as defining a probability distribution $p_{\mathcal{N}}(\mathbf{Y} \mid \mathbf{X}, \boldsymbol{\theta})$ over outputs $\mathbf{Y}$ for the given input $\mathbf{X}$ and the given network parameters $\boldsymbol{\theta}$. The goal of training is then to find parameters that maximize the likelihood $p_{\mathcal{N}}(\mathbf{Y}^* \mid \mathbf{X}, \boldsymbol{\theta})$ of the training targets under this model (maximum likelihood learning). Training can again be performed by gradient descent on an equivalent error function that is usually given by the negative log-likelihood $E_{\mathbf{X}, \mathbf{Y}^*}(\boldsymbol{\theta}) = -\log p_{\mathcal{N}}(\mathbf{Y}^* \mid \mathbf{X}, \boldsymbol{\theta})$.

Going one step further in this reasoning, a full Bayesian treatment adds prior beliefs about the network parameters through a prior distribution $p_{\mathcal{S}}(\boldsymbol{\theta})$ (we term this distribution the structural prior for reasons that will become clear below) over parameter values $\boldsymbol{\theta}$ and the training goal is formulated via the posterior distribution over parameters $\boldsymbol{\theta}$. The training goal that we consider in this article is to produce sample parameter vectors which have a high probability under the posterior distribution $p^*(\boldsymbol{\theta} \mid \mathbf{X}, \mathbf{Y}^*) \propto p_{\mathcal{S}}(\boldsymbol{\theta}) \cdot p_{\mathcal{N}}(\mathbf{Y}^* \mid \mathbf{X}, \boldsymbol{\theta})$. More generally, we are interested in a target distribution $p^*(\boldsymbol{\theta}) \propto p^*(\boldsymbol{\theta} \mid \mathbf{X}, \mathbf{Y}^*)^{\frac{1}{T}}$ that is a tempered version of the posterior where $T$ is a temperature parameter. For $T = 1$ we recover the posterior distribution, for $T > 1$ the peaks of the posterior are flattened, and for $T < 1$ the distribution is sharpened, leading to higher probabilities for parameter settings with better performance.

This training goal was explored by Welling & Teh (2011), Chen et al. (2016), and Kappel et al. (2015) where it was shown that gradient descent in combination with stochastic weight updates performs Markov Chain Monte Carlo (MCMC) sampling from the posterior distribution. In this paper we extend these results by (a) allowing the algorithm also to sample the network structure, and (b) including a hard posterior constraint on the total number of connections during the sampling process. We define the training goal as follows:

$$\text{produce samples } \boldsymbol{\theta} \text{ with high probability in } \ p^*(\boldsymbol{\theta}) \ = \ \begin{cases} 0 \text{ if } \boldsymbol{\theta} \text{ violates the constraint} \\ \frac{1}{\mathcal{Z}} p^*(\boldsymbol{\theta} \,|\, \mathbf{X}, \mathbf{Y}^*)^{\frac{1}{T}} \text{ otherwise,} \end{cases} \tag{1}$$

where $\mathcal{Z}$ is a normalizing constant. The emerging learning dynamics jointly samples from a posterior distribution over network parameters $\boldsymbol{\theta}$ and constrained network architectures. In the next section we introduce the algorithm and in Section 4 we discuss the theoretical guarantees.

**The DEEP R algorithm:** In many situations, network connectivity is strictly limited during training, for instance because of hardware memory limitations. Then the limiting factor for a training algorithm is the maximal connectivity ever needed during training. DEEP R guarantees such a hard limit. DEEP R achieves the learning goal (1) on *network configurations*, that is, it not only samples the network weights and biases, but also the connectivity under the given constraints. This is achieved by introducing the following mapping from network parameters $\boldsymbol{\theta}$ to network weights $\mathbf{w}$:

A connection parameter $\theta_k$ and a constant sign $s_k \in \{-1, 1\}$ are assigned to each connection $k$. If $\theta_k$ is negative, we say that the connection $k$ is *dormant*, and the corresponding weight is $w_k = 0$. Otherwise, the connection is considered *active*, and the corresponding weight is $w_k = s_k \theta_k$. Hence, each $\theta_k$ encodes (a) whether the connection is active in the network, and (b) the weight of the connection if it is active. Note that we use here a single index $k$ for each connection / weight instead of the more usual double index that defines the sending and receiving neuron. This connection-centric indexing is more natural for our rewiring algorithms where the connections are in the focus rather than the neurons. Using this mapping, sampling from the posterior over $\boldsymbol{\theta}$ is equivalent to sampling from the posterior over network configurations, that is, the network connectivity structure and the network weights.

---

**1** **for** $i$ *in* $[1, N_{iterations}]$ **do**
**2**      **for** *all active connections* $k$ $(\theta_k \geq 0)$ **do**
**3**          $\theta_k \leftarrow \theta_k - \eta \frac{\partial}{\partial \theta_k} E_{\mathbf{X}, \mathbf{Y}^*}(\boldsymbol{\theta}) - \eta\alpha + \sqrt{2\eta T} \ \nu_k$;
**4**          **if** $\theta_k < 0$ **then** set connection $k$ dormant ;
**5**      **end**
**6**      **while** *number of active connections lower than* $K$ **do**
**7**          select a dormant connection $k'$ with uniform probability and activate it;
**8**          $\theta_{k'} \leftarrow 0$
**9**      **end**
**10** **end**

**Algorithm 1:** Pseudo code of the DEEP R algorithm. $\nu_k$ is sampled from a zero-mean Gaussian of unit variance independently for each active and each update step. Note that the gradient of the error $E_{\mathbf{X}, \mathbf{Y}^*}(\boldsymbol{\theta})$ is computed by backpropagation over a mini-batch in practice.

---

DEEP R is defined in Algorithm 1. Gradient updates are performed only on parameters of active connections (line 3). The derivatives of the error function $\frac{\partial}{\partial \theta_k} E_{\mathbf{X}, \mathbf{Y}^*}(\boldsymbol{\theta})$ can be computed in the usual way, most commonly with the backpropagation algorithm. Since we consider only classification problems in this article, we used the cross-entropy error for the experiments in this article. The third term in line 3 $(-\eta\alpha)$ is an $\ell_1$ regularization term, but other regularizers could be used as well.

A conceptual difference to gradient descent is introduced via the last term in line 3. Here, noise $\sqrt{2\eta T} \ \nu_k$ is added to the update, where the temperature parameter $T$ controls the amount of noise and $\nu_k$ is sampled from a zero-mean Gaussian of unit variance independently for each parameter and each update step. The last term alone would implement a random walk in parameter space. Hence, the whole line 3 of the algorithm implements a combination of gradient descent on the regularized error function with a random walk. Our theoretical analysis shows that this random walk behavior

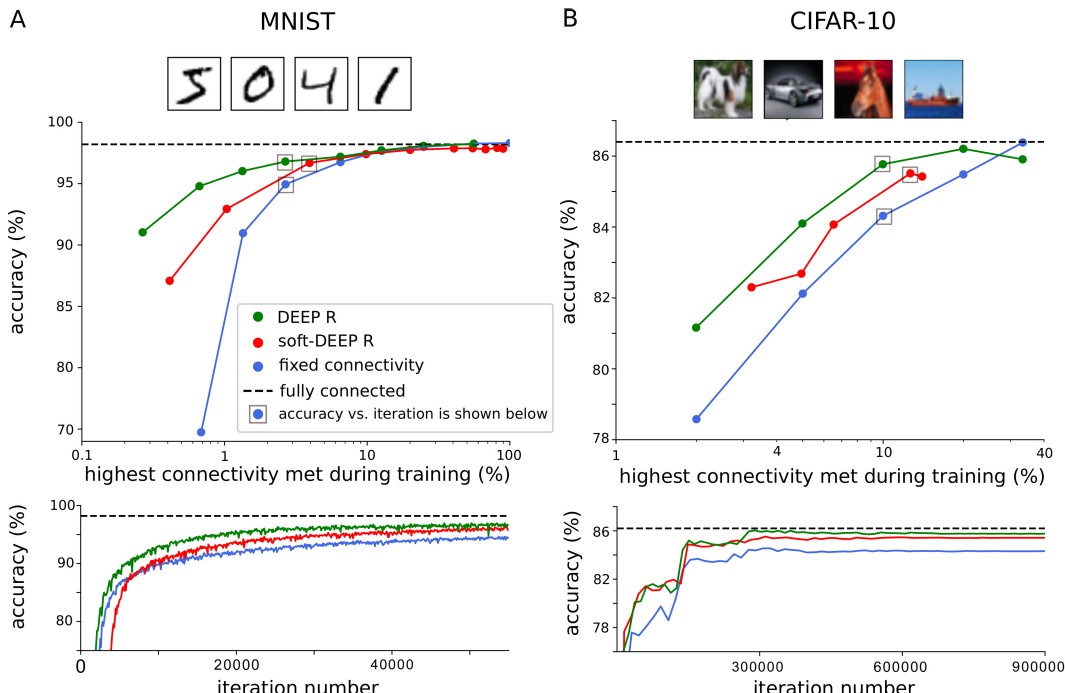

Figure 1: **Visual pattern recognition with sparse networks during training.** Sample training images (top), test classification accuracy after training for various connectivity levels (middle) and example test accuracy evolution during training (bottom) for a standard feed forward network trained on MNIST (**A**) and a CNN trained on CIFAR-10 (**B**). Accuracies are shown for various algorithms. Green: DEEP R; red: soft-DEEP R; blue: SGD with initially fixed sparse connectivity; dashed gray: SGD, fully connected. Since soft-DEEP R does not guarantee a strict upper bound on the connectivity, accuracies are plotted against the highest connectivity ever met during training (middle panels). Iteration number refers to the number of parameter updates during training.

has an important functional consequence, see the paragraph after the next for a discussion on the theoretical properties of DEEP R.

The rewiring aspect of the algorithm is captured in lines 4 and 6–9 in Algorithm (1). Whenever a parameter $\theta_k$ becomes smaller than 0, the connection is set dormant, i.e., it is deleted from the network and no longer considered for updates (line 4). For each connection that was set to the dormant state, a new connection $k'$ is chosen randomly from the uniform distribution over dormant connections, $k'$ is activated and its parameter is initialized to 0. This rewiring strategy (a) ensures that exactly $K$ connections are active at any time during training (one initializes the network with $K$ active connections), and (b) that dormant connections do not need any computational demands except for drawing connections to be activated. Note that for sparse networks, it is efficient to keep only a list of active connections and none for the dormant connections. Then, one can efficiently draw connections from the whole set of possible connections and reject those that are already active.

## 3 EXPERIMENTS

**Rewiring in fully connected and in convolutional networks:** We first tested the performance of DEEP R on MNIST and CIFAR-10. For MNIST, we considered a fully connected feed-forward network used in Han et al. (2015b) to benchmark pruning algorithms. It has two hidden layers of 300 and 100 neurons respectively and a 10-fold softmax output layer. On the CIFAR-10 dataset, we used a convolutional neural network (CNN) with two convolutional followed by two fully connected layers. For reproducibility purposes the network architecture and all parameters of this CNN were taken from the official tutorial of Tensorflow. On CIFAR-10, we used a decreasing learning rate

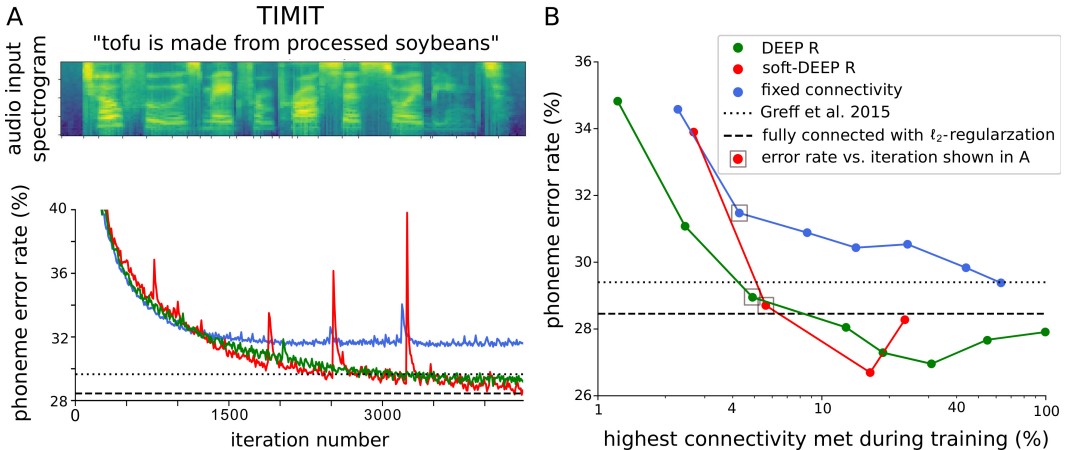

Figure 2: **Rewiring in recurrent neural networks.** Network performance for one example run (**A**) and at various connectivity levels (**B**) as in Fig. 1 for an LSTM network trained on the TIMIT dataset with DEEP R (green), soft-DEEP R (red) and a network with fixed random connectivity (blue). Dotted line: fully connected LSTM trained without regularization as reported in Greff et al. (2017). Thick dotted line: fully connected LSTM with $\ell_2$ regularization.

and a cooling schedule to reduce the temperature parameter $T$ over iterations (see Appendix A for details on all experiments).

For each task, we performed four training sessions. First, we trained a network with DEEP R. In the CNN, the first convolutional layer was kept fully connected while we allowed rewiring of the second convolutional layer. Second, we tested another algorithm, soft-DEEP R, which is a simplified version of DEEP R that does however not guarantee a strict connectivity constraint (see Section 4 for a description). Third, we trained a network in the standard manner without any rewiring or pruning to obtain a baseline performance. Finally, we trained a network with a connectivity that was randomly chosen before training and kept fixed during the optimization. The connectivity was however not completely random. Rather each layer received a number of connections that was the same as the number found by soft-DEEP R. The performance of this network is expected to be much better than a network where all layers are treated equally.

Fig. 1 shows the performance of these algorithms on MNIST (panel A) and on CIFAR-10 (panel B). DEEP R reaches a classification accuracy of 96.2 % when constrained to 1.3 % connectivity. To evaluate precisely the accuracy that is reachable with 1.0 % connectivity, we did an additional experiment where we doubled the number of training epochs. DEEP R reached a classification accuracy of 96.3% (less than 2 % drop in comparison to the fully connected baseline). Training on fixed random connectivity performed surprisingly well for connectivities around 10 %, possibly due to the large redundancy in the MNIST images. Soft-DEEP R does not guarantee a strict upper bound on the network connectivity. When considering the maximum connectivity ever seen during training, soft-DEEP R performed consistently worse than DEEP R for networks where this maximum connectivity was low. On CIFAR-10, the classification accuracy of DEEP R was 84.1 % at a connectivity level of 5 %. The performance of DEEP R at 20 % connectivity was close to the performance of the fully connected network.

To study the rewiring properties of DEEP R, we monitored the number of newly activated connections per iteration (i.e., connections that changed their status from dormant to active in that iteration). We found that after an initial transient, the number of newly activated connections converged to a stable value and remained stable even after network performance has converged, see Appendix B.

**Rewiring in recurrent neural networks:**    In order to test the generality of our rewiring approach, we also considered the training of recurrent neural networks with backpropagation through time (BPTT). Recurrent networks are quite different from their feed forward counterparts in terms of their dynamics. In particular, they are potentially unstable due to recurrent loops in inference and training signals. As a test bed, we considered an LSTM network trained on the TIMIT data set. In our

rewiring algorithms, all connections were potentially available for rewiring, including connections to gating units. From the TIMIT audio data, MFCC coefficients and their temporal derivatives were computed and fed into a bi-directional LSTMs with a single recurrent layer of 200 cells followed by a softmax to generate the phoneme likelihood (Graves & Schmidhuber, 2005), see Appendix A.

We considered as first baseline a fully connected LSTM with standard BPTT without regularization as the training algorithm. This algorithm performed similarly as the one described in Greff et al. (2017). It turned out however that performance could be significantly improved by including a regularizer in the training objective. We therefore considered the same setup with $\ell_2$ regularization (cross-validated). This setup achieved a phoneme error rate of 28.3 %. We note that better results have been reported in the literature using the CTC cost function and deeper networks (Graves et al., 2013). For the sake of easy comparison however, we sticked here to the much simpler setup with a medium-sized network and the standard cross-entropy error function.

We found that connectivity can be reduced significantly in this setup with our algorithms, see Fig. 2. Both algorithms, DEEP R and soft-DEEP R, performed even slightly better than the fully connected baseline at connectivities around 10 %, probably due to generalization issues. DEEP R outperformed soft-DEEP R at very low connectivities and it outperformed BPTT with fixed random connectivity consistently at any connectivity level considered.

**Comparison to algorithms that cannot be run on very sparse networks:** We wondered how much performance is lost when a strict connectivity constraint has to be taken into account during training as compared to pruning algorithms that only achieve sparse networks after training. To this end, we compared the performance of DEEP R and soft-DEEP R to recently proposed pruning algorithms: $\ell_1$-shrinkage (Tibshirani, 1996; Collins & Kohli, 2014) and the pruning algorithm proposed by Han et al. (2015b). $\ell_1$-shrinkage uses simple $\ell_1$-norm regularization and finds network solutions with a connectivity that is comparable to the state of the art (Collins & Kohli, 2014; Yu et al., 2012). We chose this one since it is relatively close to DEEP R with the difference that it does not implement rewiring. The pruning algorithm from Han et al. (2015b) is more complex and uses a projection of network weights on a $\ell_0$ constraint. Both algorithms prune connections starting from the fully connected network. The hyper-parameters such as learning rate, layer size, and weight decay coefficients were kept the same in all experiments. We validated by an extensive parameter search that these settings were good settings for the comparison algorithms, see Appendix A.

Results for the same setups as considered above (MNIST, CIFAR-10, TIMIT) are shown in Fig. 3. Despite the strict connectivity constraints, DEEP R and soft-DEEP R performed slightly better than the unconstrained pruning algorithms on CIFAR-10 and TIMIT at all connectivity levels considered. On MNIST, pruning was slightly better for larger connectivities. On MNIST and TIMIT, pruning and $\ell_1$-shrinkage failed completely for very low connectivities while rewiring with DEEP R or soft-DEEP R still produced reasonable networks in this case.

One interesting observation can be made for the error rate evolution of the LSTM on TIMIT (Fig. 3D). Here, both $\ell_1$-shrinkage and pruning induced large sudden increases of the error rate, possibly due to instabilities induced by parameter changes in the recurrent network. In contrast, we observed only small glitches of this type in DEEP R. This indicates that sparsification of network connectivity is harder in recurrent networks due to potential instabilities, and that DEEP R is better suited to avoid such instabilities. The reason for this advantage of DEEP R is however not clear.

**Transfer learning is supported by DEEP R:** If the temperature parameter $T$ is kept constant during training, the proposed rewiring algorithms do not converge to a static solution but explore continuously the posterior distribution of network configurations. As a consequence, rewiring is expected to adapt to changes in the task in an on line manner. If the task demands change in an online learning setup, one may hope that a transfer of invariant aspects of the tasks occurs such that these aspects can be utilized for faster convergence on later tasks (transfer learning). To verify this hypothesis, we performed one experiment on the MNIST dataset where the class to which each output neuron should respond to was changed after each training epoch (class-shuffled MNIST task). Fig. 4A shows the performance of a network trained with DEEP R in the class-shuffled MNIST task. One can observe that performance recovered after each shuffling of the target classes. More importantly, we found a clear trend of increasing classification accuracy even across shuffles. This indicates a form of transfer learning in the network such that information about the previous tasks

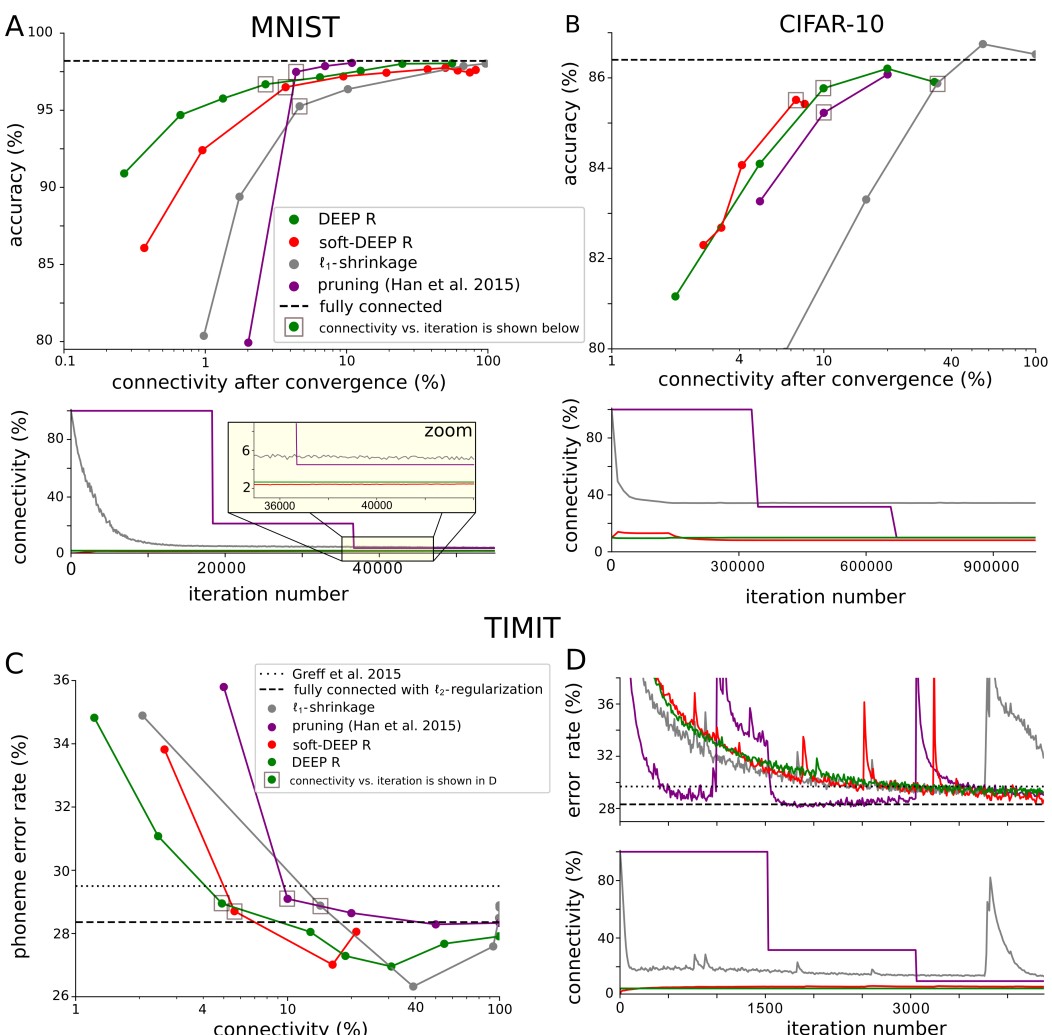

Figure 3: **Efficient network solutions under strict sparsity constraints.** Accuracy and connectivity obtained by DEEP R and soft-DEEP R in comparison to those achieved by pruning (Han et al., 2015b) and $\ell_1$-shrinkage (Tibshirani, 1996; Collins & Kohli, 2014). **A, B)** Accuracy against the connectivity for MNIST (A) and CIFAR-10 (B). For each algorithm, one network with a decent compromise between accuracy and sparsity is chosen (small gray boxes) and its connectivity across training iterations is shown below. **C)** Performance on the TIMIT dataset. **D)** Phoneme error rates and connectivities across iteration number for representative training sessions.

(i.e., the previous target-shuffled MNIST instances) was preserved in the network and utilized in the following instances. We hypothesized for the reason of this transfer that early layers developed features that were invariant to the target shuffling and did not need to be re-learned in later task instances. To verify this hypothesis, we computed the following two quantities. First, in order to quantify the speed of parameter dynamics in different layers, we computed the correlation between the layer weight matrices of two subsequent training epoch (Fig. 4B). Second, in order to quantify the speed of change of network dynamics in different layers, we computed the correlation between the neuron outputs of a layer in subsequent epochs (Fig. 4C). We found that the correlation between weights and layer outputs increased across training epochs and were significantly larger in early layers. This supports the hypothesis that early network layers learned features invariant to the shuffled coding convention of the output layer.

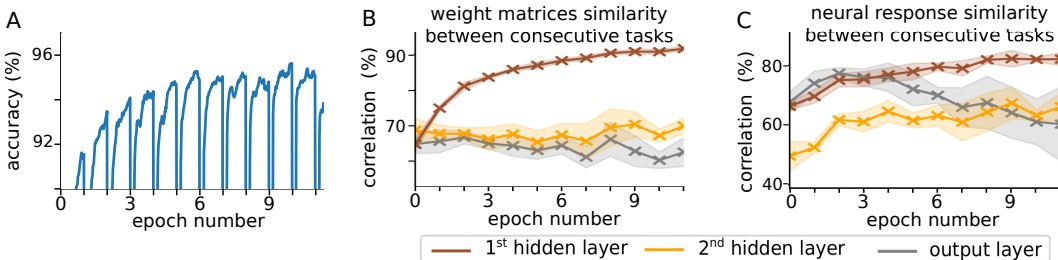

Figure 4: **Transfer learning with DEEP R.** The target labels of the MNIST data set were shuffled after every epoch. **A)** Network accuracy vs. training epoch. The increase of network performance across tasks (epochs) indicates a transfer of knowledge between tasks. **B)** Correlation between weight matrices of subsequent epochs for each network layer. **C)** Correlation between neural activity vectors of subsequent epochs for each network layer. The transfer is most visible in the first hidden layer, since weights and outputs of this layer are correlated across tasks. Shaded areas in **B)** and **C)** represent standard deviation across 5 random seeds, influencing network initialization, noisy parameter updates, and shuffling of the outputs.

## 4 Convergence properties of DEEP R and soft-DEEP R

The theoretical analysis of DEEP R is somewhat involved due to the implemented hard constraints. We therefore first introduce and discuss here another algorithm, soft-DEEP R where the theoretical treatment of convergence is more straight forward. In contrast to standard gradient-based algorithms, this convergence is not a convergence to a particular parameter vector, but a convergence to the target distribution over network configurations.

---

1 **for** $i$ *in* $[1, N_{iterations}]$ **do**
2     **for** *all active connections* $k$ $(\theta_k \geq 0)$ **do**
3         $\theta_k \leftarrow \theta_k - \eta \frac{\partial}{\partial \theta_k} E_{\mathbf{X}, \mathbf{Y}^*}(\boldsymbol{\theta}) - \eta\alpha + \sqrt{2\eta T}\, \nu_k;$
4         **if** $\theta_k < 0$ **then** set connection $k$ dormant ;
5     **end**
6     **for** *all dormant connections* $k$ $(\theta_k < 0)$ **do**
7         $\theta_k \leftarrow \theta_k + \sqrt{2\eta T}\, \nu_k;$
8         $\theta_k \leftarrow \max\{\theta_k, \theta_{\min}\};$
9         **if** $\theta_k \geq 0$ **then** set connection $k$ active ;
10     **end**
11 **end**

**Algorithm 2:** Pseudo code of the soft-DEEP R algorithm. $\theta_{\min} < 0$ is a constant that defines a lower boundary for negative $\theta_k$s.

**Convergence properties of soft-DEEP R:**    The soft-DEEP R algorithm is given in Algorithm 2. Note that the updates for active connections are the same as for DEEP R (line 3). Also the mapping from parameters $\theta_k$ to weights $w_k$ is the same as in DEEP R. The main conceptual difference to DEEP R is that connection parameters continue their random walk when dormant (line 7). Due to this random walk, connections will be re-activated at random times when they cross zero. Therefore, soft-DEEP R does not impose a hard constraint on network connectivity but rather uses the $\ell_1$ norm regularization to impose a soft-constraint.

Since dormant connections have to be simulated, this algorithm is computationally inefficient for sparse networks. An approximation could be used where silent connections are re-activated at a constant rate, leading to an algorithm very similar to DEEP R. DEEP R adds to that the additional feature of a strict connectivity constraint.

The central result for soft-DEEP R has been proven in the context of spiking neural networks in (Kappel et al., 2015) in order to understand rewiring in the brain from a functional perspective. The same theory however also applies to standard deep neural networks. To be able to apply standard

mathematical tools, we consider parameter dynamics in continuous time. In particular, consider the following stochastic differential equation (SDE)

$$d\theta_k \;=\; \beta \; \frac{\partial}{\partial \theta_k} \log p^*(\boldsymbol{\theta}|\mathbf{X}, \mathbf{Y}^*) \Big|_{\boldsymbol{\theta}^t} dt \;+\; \sqrt{2\beta T}\, d\mathcal{W}_k \;, \tag{2}$$

where $\beta$ is the equivalent to the learning rate and $\frac{\partial}{\partial \theta_k} \log p^*(\boldsymbol{\theta}|\mathbf{X}, \mathbf{Y}^*) \Big|_{\boldsymbol{\theta}^t}$ denotes the gradient of the log parameter posterior evaluated at the parameter vector $\boldsymbol{\theta}^t$ at time $t$. The term $d\mathcal{W}_k$ denotes the infinitesimal updates of a standard Wiener process. This SDE describes gradient ascent on the log posterior combined with a random walk in parameter space. We show in Appendix C that the unique stationary distribution of this parameter dynamics is given by

$$p^*(\boldsymbol{\theta}) = \frac{1}{\mathcal{Z}} p^*(\boldsymbol{\theta} \,|\, \mathbf{X}, \mathbf{Y}^*)^{\frac{1}{T}} \;. \tag{3}$$

Since we considered classification tasks in this article, we interpret the network output as a multi-nomial distribution over class labels. Then, the derivative of the log likelihood is equivalent to the derivative of the negative cross-entropy error. Together with an $\ell_1$ regularization term for the prior, and after discretization of time, we obtain the update of line 3 in Algorithm 2 for non-negative parameters. For negative parameters, the first term in Eq. (2) vanishes since the network weight is constant zero there. This leads to the update in line 7. Note that we introduced a reflecting boundary at $\theta_{\min} < 0$ in the practical algorithm to avoid divergence of parameters (line 8).

**Convergence properties of DEEP R:**  A detailed analysis of the stochastic process that underlies the algorithm is provided in Appendix D. Here we summarize the main findings. Each iteration of DEEP R in Algorithm 1 consists of two parts: In the first part (lines 2-5) all connections that are currently active are advanced, while keeping the other parameters at 0. In the second part (lines 6-9) the connections that became dormant during the first step are randomly replenished.

To describe the connectivity constraint over connections we introduce the binary constraint vector $\mathbf{c}$ which represents the set of active connections, i.e., element $c_k$ of $\mathbf{c}$ is 1 if connection $k$ is allowed to be active and zero else. In Theorem 2 of Appendix D, we link DEEP R to a compound Markov chain operator that simultaneously updates the parameters $\boldsymbol{\theta}$ according to the soft-DEEP R dynamics under the constraint $\mathbf{c}$ and the constraint vector $\mathbf{c}$ itself. The stationary distribution of this Markov chain is given by the joint probability

$$p^*(\boldsymbol{\theta}, \mathbf{c}) \;\propto\; p^*(\boldsymbol{\theta})\, \mathcal{C}(\boldsymbol{\theta}, \mathbf{c})\, p_{\mathcal{C}}(\mathbf{c}) \;, \tag{4}$$

where $\mathcal{C}(\boldsymbol{\theta}, \mathbf{c})$ is a binary function that indicates compatibility of $\boldsymbol{\theta}$ with the constraint $\mathbf{c}$ and $p^*(\boldsymbol{\theta})$ is the tempered posterior of Eq. (3) which is left stationary by soft-DEEP R in the absence of con-straints. $p_{\mathcal{C}}(\mathbf{c})$ in Eq. (4) is a uniform prior over all connectivity constraints with exactly $K$ synapses that are allowed to be active. By marginalizing over $\mathbf{c}$, we obtain that the posterior distribution of DEEP R is identical to that of soft-DEEP R if the constraint on the connectivity is fulfilled. By marginalizing over $\boldsymbol{\theta}$, we obtain that the probability of sampling a network architecture (i.e. a con-nectivity constraint $\mathbf{c}$) with DEEP R and soft-DEEP R are proportional to one another. The only difference is that DEEP R exclusively visits architectures with $K$ active connections (see equation (39) in Appendix D for details).

In other words, DEEP R solves a constraint optimization problem by sampling parameter vectors $\boldsymbol{\theta}$ with high performance within the space of constrained connectivities. The algorithm will there-fore spend most time in network configurations where the connectivity supports the desired network function, such that, connections with large support under the objective function (1) will be main-tained active with high probability, while other connections are randomly tested and discarded if found not useful.

## 5  DISCUSSION

**Related Work:** de Freitas et al. (2000) considered sequential Monte Carlo sampling to train neural networks by combining stochastic weight updates with gradient updates. Stochastic gradient up-dates in mini-batch learning was considered in Welling & Teh (2011), where also a link to the true

posterior distribution was established. Chen et al. (2016) proposed a momentum scheme and temperature annealing (for the temperature $T$ in our notation) for stochastic gradient updates, leading to a stochastic optimization method. DEEP R extends this approach by using stochastic gradient Monte Carlo sampling not only for parameter updates but also to sample the connectivity of the network. In addition, the posterior in DEEP R is subject to a hard constraint on the network architecture. In this sense, DEEP R performs constrained sampling, or constrained stochastic optimization if the temperature is annealed. Patterson & Teh (2013) considered the problem of stochastic gradient dynamics constrained to the probability simplex. The methods considered there are however not readily applicable to the problem of constraints on the connection matrix considered here. Additionally, we show that a correct sampler can be constructed that does not simulate dormant connections. This sampler is efficient for sparse connection matrices. Thus, we developed a novel method, random reintroduction of connections, and analyzed its convergence properties (see Theorem 2 in Appendix D).

**Conclusions:** We have presented a method for modifying backprop and backprop-through-time so that not only the weights of connections, but also the connectivity graph is simultaneously optimized during training. This can be achieved while staying always within a given bound on the total number of connections. When the absolute value of a weight is moved by backprop through $0$, it becomes a weight with the opposite sign. In contrast, in DEEP R a connection vanishes in this case (more precisely: becomes dormant), and a randomly drawn other connection is tried out by the algorithm. This setup requires that, like in neurobiology, the sign of a weight does not change during learning. Another essential ingredient of DEEP R is that it superimposes the gradient-driven dynamics of each weight with a random walk. This feature can be viewed as another inspiration from neurobiology (Mongillo et al., 2017). An important property of DEEP R is that — in spite of its stochastic ingredient — its overall learning dynamics remains theoretically tractable: Not as gradient descent in the usual sense, but as convergence to a stationary distribution of network configurations which assigns the largest probabilities to the best-performing network configurations. An automatic benefit of this ongoing stochastic parameter dynamics is that the training process immediately adjusts to changes in the task, while simultaneously transferring previously gained competences of the network (see Fig. 4).

**Acknowledgements** Written under partial support by the Human Brain Project of the European Union #720270, and the Austrian Science Fund (FWF): I 3251-N33. We thank Franz Pernkopf and Matthias Zöhrer for useful comments regarding the TIMIT experiment.

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

## A  METHODS

Implementations of DEEP R are freely available at github.com/guillaumeBellec/deep_rewiring.

**Choosing hyper-parameters for DEEP R:**  The learning rate $\eta$ is defined for each task independently (see task descriptions below). Considering that the number of active connections is given as a constraint, the remaining hyper parameters are the regularization coefficient $\alpha$ and the temperature $T$. We found that the performance of DEEP R does not depend strongly on the temperature $T$. Yet, the choice of $\alpha$ has to be done more carefully. For each dataset there was an ideal value of $\alpha$: one order of magnitude higher or lower typically lead to a substantial loss of accuracy.

In MNIST, 96.3% accuracy under the constraint of 1% connectivity was achieved with $\alpha = 10^{-4}$ and $T$ chosen so that $T = \frac{\eta}{2}10^{-12}$. In TIMIT, $\alpha = 0.03$ and $T = 0$ (higher values of $T$ could improve the performance slightly but it did not seem very significant). In CIFAR-10 a different $\alpha$ was assigned to each connectivity matrix. To reach 84.1% accuracy with 5% connectivity we used in each layer from input to output $\alpha = [0, 10^{-7}, 10^{-6}, 10^{-9}, 0]$. The temperature is initialized with $T = \eta\frac{\alpha^2}{18}$ and decays with the learning rate (see paragraph of the methods about CIFAR-10).

**Choosing hyper-parameters for soft-DEEP R:**  The main difference between soft-DEEP R and DEEP R is that the connectivity is not given as a global constraint. This is a considerable drawback if one has strict constraint due to hardware limitation but it is also an advantage if one simply wants to generate very sparse network solutions without having a clear idea on the connectivities that are reachable for the task and architecture considered.

In any cases, the performance depends on the choice of hyper-parameters $\alpha$, $T$ and $\theta_{min}$, but also - unlike in DEEP R - these hyper parameters have inter-dependent relationships that one cannot ignore (as for DEEP R, the learning rate $\eta$ is defined for each task independently). The reason why soft-DEEP R depends more on the temperature is that the rate of re-activation of connections is driven by the amplitude of the noise whereas they are decoupled in DEEP R. To summarize the results of an exhaustive parameter search, we found that $\sqrt{2T\eta}$ should ideally be slightly below $\alpha$. In general high $\theta_{min}$ leads to high performance but it also defines an approximate lower bound on the smallest reachable connectivity. This lower bound can be estimated by computing analytically the stationary distribution under rough approximations and the assumption that the gradient of the likelihood is zero. If $p_{min}$ is the targeted lower connectivity bound, one needs $\theta_{min} \approx -\frac{T(1-p_{min})}{\alpha p_{min}}$.

For MNIST we used $\alpha = 10^{-5}$ and $T = \eta\frac{\alpha^2}{18}$ for all data points in Fig. 1 panel A and a range of values of $\theta_{min}$ to scope across different ranges of connectivity lower bounds. In TIMIT and CIFAR-10 we used a simpler strategy which lead to a similar outcome, we fixed the relationships: $\alpha = 3\sqrt{2\frac{T}{\eta}} = \frac{-1}{3}\theta_{min}$ and we varied only $\alpha$ to produce the solutions shown in Fig. 1 panel B and Fig. 2.

**Re-implementing pruning and $\ell_1$-shrinkage:**  To implement $\ell_1$-shrinkage (Tibshirani, 1996; Collins & Kohli, 2014), we applied the $\ell_1$-shrinkage operator $\theta \leftarrow \text{relu}\left(|\theta| - \eta\alpha\right)\text{sign}(\theta)$ after each gradient descent iteration. The performance of the algorithm is evaluated for different $\alpha$ varying on a logarithmic scale to privilege a sparse connectivity or a high accuracy. For instance for MNIST in Figure 3.A we used $\alpha$ of the form $10^{-\frac{n}{2}}$ with $n$ going from 4 to 12. The optimal parameter was $n = 9$.

We implemented the pruning described in Han et al. (2015b). This algorithm uses several phases: training - pruning - training, or one can also add another pruning iteration: training - pruning - training - pruning - training. We went for the latter because it increased performance. Each "training" phase is a complete training of the neural network with $\ell_2$-regularization[1]. At each "pruning" phase, the standard deviation of weights within a weight matrix $w_{\text{std}}$ is computed and all active weights with absolute values smaller than $qw_{\text{std}}$ are pruned ($q$ is called the quality parameter). Grid search is

---

[1]To be fair with other algorithms, we did not allocate three times more training time to pruning, each "training" phase was performed for a third of the total number of epochs which was chosen much larger than necessary.

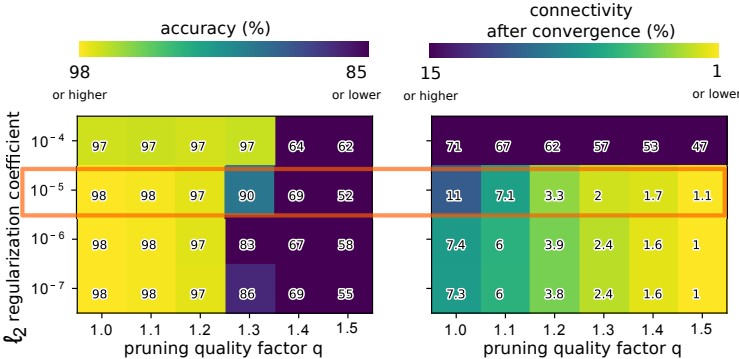

Figure 5: **Hyper-parameter search for the pruning algorithm according to Han et al. (2015b).**
Each point of the grid represents a weight decay coefficient – quality factor pair. The number and the
color indicate the performance in terms of accuracy (left) or connectivity (right). The red rectangle
indicates the data points that were used in Fig. 3A.

used to optimize the $\ell_2$-regularization coefficient and quality parameter. The results for MNIST are
reported in Figure 5.

**MNIST:** We used a standard feed forward network architecture with two hidden layers with 200
neurons each and rectified linear activation functions followed by a 10-fold softmax output. For all
algorithms we used a learning rate of 0.05 and a batch size of 10 with standard stochastic gradient
descent. Learning stopped after 10 epochs. All reported performances in this article are based on
the classification error on the MNIST test set.

**CIFAR-10:** The official tutorial for convolutional networks of tensorflow[2] is used as a reference
implementation. Its performance out-of-the-box provides the fully connected baseline. We used the
values given in the tutorial for the hyper-parameters in all algorithms. In particular the layer-specific
weight decay coefficients that interact with our algorithms were chosen from the tutorial for DEEP
R, soft-DEEP R, pruning, and $\ell_1$-shrinkage.

In the fully connected baseline implementation, standard stochastic gradient descent was used with
a decreasing learning rate initialized to 1 and decayed by a factor 0.1 every 350 epochs. Training
was performed for one million iterations for all algorithms. For soft-DEEP R, which includes a
temperature parameter, keeping a high temperature as the weight decays was increasing the rate
of re-activation of connections. Even if intermediate solutions were rather sparse and efficient the
solutions after convergence were always dense. Therefore, the weight decay was accompanied by
annealing of the temperature $T$. This was done by setting the temperature to be proportional to the
decaying $\eta$. This annealing was used for DEEP R and soft-DEEP R.

**TIMIT:** The TIMIT dataset was preprocessed and the LSTM architecture was chosen to reproduce
the results from Greff et al. (2017). Input time series were formed by 12 MFCC coefficients and the
log energy computed over each time frame. The inputs were then expanded with their first and
second temporal derivatives. There are 61 different phonemes annotated in the TIMIT dataset, to
report an error rate that is comparable to the literature we performed a standard grouping of the
phonemes to generate 39 output classes (Lee & Hon, 1989; Graves et al., 2013; Greff et al., 2017).
As usual, the dialect specific sentences were excluded (SA files). The phoneme error rate was
computed as the proportion of misclassified frames.

A validation set and early stopping were necessary to train a network with dense connectivity matrix
on TIMIT because the performance was sometimes unstable and it suddenly dropped during training
as seen in Fig. 3D for $\ell_1$-shrinkage. Therefore a validation set was defined by randomly selecting
5% of the training utterances. All algorithms were trained for 40 epochs and the reported test error
rate is the one at minimal validation error.

---

[2]TensorFlow version 1.3: www.tensorflow.org/tutorials/deep_cnn

To accelerate the training in comparison the reference from Greff et al. (2017) we used mini-batches of size 32 and the ADAM optimizer (Kingma & Ba (2014)). This was also an opportunity to test the performance of DEEP R and soft-DEEP R with such a variant of gradient descent. The learning rate was set to 0.01 and we kept the default momentum parameters of ADAM, yet we found that changing the $\epsilon$ parameter (as defined in Kingma & Ba (2014)) from $10^{-8}$ to $10^{-4}$ improved the stability of fully connected networks during training in this recurrent setup. As we could not find a reference that implemented $\ell_1$-shrinkage in combination with ADAM, we simply applied the shrinkage operator after each iteration of ADAM which might not be the ideal choice in theory. It worked well in practice as the minimal error rate was reached with this setup. The same type of $\ell_1$ regularization in combination with ADAM was used for DEEP R and soft-DEEP R which lead to very sparse and efficient network solutions.

**Initialization of connectivity matrices:** We found that the performance of the networks depended strongly on the initial connectivity. Therefore, we followed the following heuristics to generate initial connectivity for DEEP R, soft-DEEP R and the control setup with fixed connectivity.

First, for the connectivity matrix of each individual layer, the zero entries were chosen with uniform probability. Second, for a given connectivity constraint we found that the learning time increased and the performance dropped if the initial connectivity matrices were not chosen carefully. Typically the performance dropped drastically if the output layer was initialized to be very sparse. Yet in most networks the number of parameters is dominated by large connectivity matrices to hidden layers. A basic rule of thumb that worked in our cases was to give an equal number of active connections to the large and intermediate weight matrices, whereas smaller ones - typically output layers - should be densely connected.

We suggest two approaches to refine this guess: One can either look at the statistics of the connectivity matrices after convergence of DEEP R or soft-DEEP R, or, if possible, the second alternative is to initialize once soft-DEEP R with a dense matrix and observe the connectivity matrix after convergence. In our experiments the connectivities after convergence were coherent with the rule of thumb described above and we did not need to pursue intensive search for ideal initial connectivity matrices.

For MNIST, the number of parameters in each layer was 235k, 30k and 1k from input to output. Using our rule of thumb, for a given global connectivity $p_0$, the layers were respectively initialized with connectivity $0.75p_0$, $2.3p_0$ and $22.8p_0$.

For CIFAR-10, the baseline network had two convolutional layers with filters of shapes $5 \times 5 \times 3 \times 64$ and $5 \times 5 \times 64 \times 64$ respectively, followed by two fully connected layer with weight matrices of shape $2304 \times 384$ and $384 \times 192$. The last layer was then projected into a softmax over 10 output classes. The numbers of parameters per connectivity matrices were therefore 5k, 102k, 885k, 738k and 2k from input to output. The connectivity matrices were initialized with connectivity $1, 8p_0, 0.8p_0, 8p_0$, and $1$.

For TIMIT, the connection matrix from the input to the hidden layer was of size $39 \times 800$, the recurrent matrix had size $200 \times 800$ and the size of the output matrix was $200 \times 39$. Each of these three connectivity matrices were initialized with a connectivity of $3p_0, p_0$, and $10p_0$ respectively.

**Initialization of weight matrices:** For CIFAR-10 the initialization of matrix coefficients was given by the reference implementation. For MNIST and TIMIT, the weight matrices were initialized with $\boldsymbol{\theta} = \frac{1}{\sqrt{n_{in}}}\mathcal{N}(0,1)\mathbf{c}$ where $n_{in}$ is the number of afferent neurons, $\mathcal{N}(0,1)$ samples from a centered gaussian with unit variance and $\mathbf{c}$ is a binary connectivity matrix.

It would not be good to initialize the parameters of all dormant connections to zero in soft-DEEP R. After a single noisy iteration, half of them would become active which would fail to initialize the network with a sparse connectivity matrix. To balance out this problem we initialized the parameters of dormant connections uniformly between the clipping value $\theta_{min}$ and zero in soft-DEEP R.

**Parameters for Figure 4** The experiment provided in Figure 4 is a variant of our MNIST experiment where the target labels were shuffled after every training epoch. To make the generalization capability of DEEP R over a small number of epochs visible, we enhanced the noise exploration by setting a batch to 1 so that the connectivity matrices were updated at every time step. Also we used

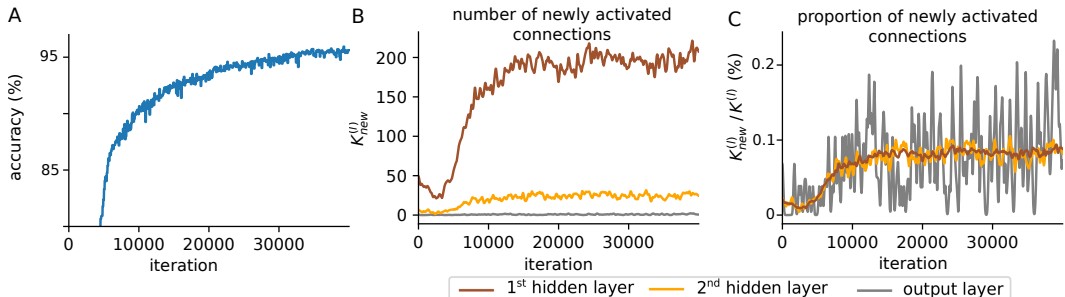

Figure 6: **Rewiring behavior of DEEP R. A)** Network performance versus training iteration (same as green line in Fig. 1A bottom, but for a network constrained to $1\%$ connectivity). **B)** Absolute number of newly activated connections $K_{\text{new}}^{(l)}$ to layer $l = 1$ (brown), $l = 2$ (orange), and the output layer ($l = 3$, gray) per iteration. Note that these layers have quite different numbers of potential connections $K^{(l)}$. **C)** Same as panel B but the number of newly activated connections are shown relative to the number of potential connections in the layer (values in panel C are smoothed with a boxcar filter over $X$ iterations).

a larger network with 400 neurons in each hidden layer. The remaining parameters were similar to those used previously: the connectivity was constrained to $1\%$ and the connectivity matrices were initialized with respective connectivities: $0.01$, $0.01$, and $0.1$. The parameters of DEEP R were set to $\eta = 0.05$, $\alpha = 10^{-5}$ and $T = \eta \frac{\alpha^2}{2}$.

## B    REWIRING DURING TRAINING ON MNIST

Fig. 6 shows the rewiring behavior of DEEP R per network layer for the feed-forward neural network trained on MNIST and the training run indicated by the small gray box around the green dot in Fig. 1A. Since it takes some iterations until the weights of connections that do not contribute to a reduction of the error are driven to $0$, the number of newly established connections $K_{\text{new}}^{(l)}$ in layer $l$ is small for all layers initially. After this initial transient, the number of newly activated connections stabilized to a value that is proportional to the total number of potential connections in the layer (Fig. 1B). DEEP R continued to rewire connections even late in the training process.

## C    DETAILS TO: CONVERGENCE PROPERTIES OF SOFT-DEEP R

Here we provide additional details on the convergence properties of the soft-DEEP R parameter update provided in Algorithm 2. We reiterate here Eq. (2):

$$d\theta_k = \beta \left. \frac{\partial}{\partial \theta_k} \log p^*(\boldsymbol{\theta}|\mathbf{X}, \mathbf{Y}^*) \right|_{\boldsymbol{\theta}^t} dt + \sqrt{2\beta T} \, d\mathcal{W}_k . \tag{5}$$

Discrete time updates can be recovered from the set of SDEs (5) by integration over a short time period $\Delta t$

$$\Delta\theta_k = \eta \frac{\partial}{\partial \theta_k} \log p^*(\boldsymbol{\theta}|\mathbf{X}, \mathbf{Y}^*) + \sqrt{2\eta T} \, \nu_k, \tag{6}$$

where the learning rate $\eta$ is given by $\eta = \beta \, \Delta t$.

We prove that the stochastic parameter dynamics Eq. (5) converges to the target distribution $p^*(\boldsymbol{\theta})$ given in Eq. (3). The proof is analogous to the derivation given in Kappel et al. (2015; 2017). We reiterate the proof here for the special case of supervised learning. The fundamental property of the synaptic sampling dynamics Eq. (5) is formalized in Theorem 1 and proven below. Before we state the theorem, we briefly discuss its statement in simple terms. Consider some initial parameter setting $\boldsymbol{\theta}^0$. Over time, the parameters change according to the dynamics (5). Since the dynamics include a noise term, the exact value of the parameters $\boldsymbol{\theta}(t)$ at some time $t > 0$ cannot be determined. However, it is possible to describe the exact distribution of parameters for each time $t$. We denote this

distribution by $p_{\text{FP}}(\boldsymbol{\theta}, t)$, where the "FP" subscript stands for "Fokker-Planck" since the evolution of this distribution is described by the Fokker-Planck equation (7) given below. Note that we make the dependence of this distribution on time explicit in this notation. It can be shown that for the dynamics (7), $p_{\text{FP}}(\boldsymbol{\theta}, t)$ converges to a well-defined and unique *stationary distribution* in the limit of large $t$. To prove the convergence to the stationary distribution we show that it is kept invariant by the set of SDEs Eq. (5) and that it can be reached from any initial condition.

We now state Theorem 1 formally. To simplify notation we drop in the following the explicit time dependence of the parameters $\boldsymbol{\theta}$.

**Theorem 1.** *Let $p^*(\boldsymbol{\theta} \,|\, \mathbf{X}, \mathbf{Y}^*)$ be a strictly positive, continuous probability distribution over parameters $\boldsymbol{\theta}$, twice continuously differentiable with respect to $\boldsymbol{\theta}$, and let $\beta > 0$. Then the set of stochastic differential equations Eq. (5) leaves the distribution $p^*(\boldsymbol{\theta})$ (3) invariant. Furthermore, $p^*(\boldsymbol{\theta})$ is the unique stationary distribution of the sampling dynamics.*

*Proof.* The stochastic differential equation Eq. (5) translates into a Fokker-Planck equation (Gardiner, 2004) that describes the evolution of the distribution over parameters $\boldsymbol{\theta}$

$$
\frac{\partial}{\partial t} p_{\text{FP}}(\boldsymbol{\theta}, t) = \sum_k -\frac{\partial}{\partial \theta_k} \left( \beta \frac{\partial}{\partial \theta_k} \log p^*(\boldsymbol{\theta} \,|\, \mathbf{X}, \mathbf{Y}^*) \right) p_{\text{FP}}(\boldsymbol{\theta}, t) + \frac{\partial^2}{\partial \theta_k^2} \left( \beta\, T\, p_{\text{FP}}(\boldsymbol{\theta}, t) \right), \quad (7)
$$

where $p_{\text{FP}}(\boldsymbol{\theta}, t)$ denotes the distribution over network parameters at time $t$. To show that $p^*(\boldsymbol{\theta})$ leaves the distribution invariant, we have to show that $\frac{\partial}{\partial t} p_{\text{FP}}(\boldsymbol{\theta}, t) = 0$ (i.e., $p_{\text{FP}}(\boldsymbol{\theta}, t)$ does not change) if we set $p_{\text{FP}}(\boldsymbol{\theta}, t)$ to $p^*(\boldsymbol{\theta})$. Plugging in the presumed stationary distribution $p^*(\boldsymbol{\theta})$ for $p_{\text{FP}}(\boldsymbol{\theta}, t)$ on the right hand side of Eq. (7), one obtains

$$
\begin{aligned}
\frac{\partial}{\partial t} p_{\text{FP}}(\boldsymbol{\theta}, t) &= \sum_k -\frac{\partial}{\partial \theta_k} \left( \beta \frac{\partial}{\partial \theta_k} \log p^*(\boldsymbol{\theta} \,|\, \mathbf{X}, \mathbf{Y}^*)\, p^*(\boldsymbol{\theta}) \right) + \frac{\partial^2}{\partial \theta_k^2} \left( \beta\, T\, p^*(\boldsymbol{\theta}) \right) \\
&= \sum_k -\frac{\partial}{\partial \theta_k} \left( \beta\, p^*(\boldsymbol{\theta}) \frac{\partial}{\partial \theta_k} \log p^*(\boldsymbol{\theta} \,|\, \mathbf{X}, \mathbf{Y}^*) \right) + \frac{\partial}{\partial \theta_k} \left( \beta\, T \frac{\partial}{\partial \theta_k} p^*(\boldsymbol{\theta}) \right) \\
&= \sum_k -\frac{\partial}{\partial \theta_k} \left( \beta\, p^*(\boldsymbol{\theta}) \frac{\partial}{\partial \theta_k} \log p^*(\boldsymbol{\theta} \,|\, \mathbf{X}, \mathbf{Y}^*) \right) + \frac{\partial}{\partial \theta_k} \left( \beta\, T\, p^*(\boldsymbol{\theta}) \frac{\partial}{\partial \theta_k} \log p^*(\boldsymbol{\theta}) \right) ,
\end{aligned}
$$

which by inserting $p^*(\boldsymbol{\theta}) = \frac{1}{\mathcal{Z}} p^*(\boldsymbol{\theta} \,|\, \mathbf{X}, \mathbf{Y}^*)^{\frac{1}{T}}$, with normalizing constant $\mathcal{Z}$, becomes

$$
\begin{aligned}
\frac{\partial}{\partial t} p_{\text{FP}}(\boldsymbol{\theta}, t) &= \frac{1}{\mathcal{Z}} \sum_k -\frac{\partial}{\partial \theta_k} \left( \beta\, p^*(\boldsymbol{\theta}) \frac{\partial}{\partial \theta_k} \log p^*(\boldsymbol{\theta} \,|\, \mathbf{X}, \mathbf{Y}^*) \right) \\
&\quad + \frac{\partial}{\partial \theta_k} \left( \beta\, T\, p^*(\boldsymbol{\theta}) \frac{1}{T} \frac{\partial}{\partial \theta_k} \log p^*(\boldsymbol{\theta} \,|\, \mathbf{X}, \mathbf{Y}^*) \right) = \sum_k 0 = 0 .
\end{aligned}
$$

This proves that $p^*(\boldsymbol{\theta})$ is a stationary distribution of the parameter sampling dynamics Eq. (5). Since $\beta$ is positive by construction, the Markov process of the SDEs (5) is ergodic and the stationary distribution is unique (see Section 5.3.3. and 3.7.2 in Gardiner (2004)).

The unique stationary distribution of Eq. (7) is given by $p^*(\boldsymbol{\theta}) = \frac{1}{\mathcal{Z}} p^*(\boldsymbol{\theta} | \mathbf{X}, \mathbf{Y}^*)^{\frac{1}{T}}$, i.e., $p^*(\boldsymbol{\theta})$ is the only solution for which $\frac{\partial}{\partial t} p_{\text{FP}}(\boldsymbol{\theta}, t)$ becomes 0, which completes the proof. $\qquad \square$

The updates of the soft-DEEP R algorithm (Algorithm 2) can be written as

$$
\Delta \theta_k = \begin{cases} \sqrt{2T\eta}\, \nu_k & \text{if } \theta_k < 0 \text{ (dormant connection)} \\ -\eta \frac{\partial}{\partial \theta_k} E_{\mathbf{X}, \mathbf{Y}^*}(\boldsymbol{\theta}) - \eta \alpha + \sqrt{2T\eta}\, \nu_k & \text{otherwise.} \end{cases} \quad (8)
$$

Eq. (8) is a special case of the general discrete parameter dynamics (6). To see this we apply Bayes' rule to expand the derivative of the log posterior into the sum of the derivatives of the prior and the likelihood:

$$
\frac{\partial}{\partial \theta_k} \log p^*(\boldsymbol{\theta} | \mathbf{X}, \mathbf{Y}^*) = \frac{\partial}{\partial \theta_k} \log p_{\mathcal{S}}(\boldsymbol{\theta}) + \frac{\partial}{\partial \theta_k} \log p_{\mathcal{N}}(\mathbf{Y}^* | \mathbf{X}, \boldsymbol{\theta}) ,
$$

such that we can rewrite Eq. (6)

$$\Delta \theta_k = \eta \left( \frac{\partial}{\partial \theta_k} \log p_{\mathcal{S}}(\boldsymbol{\theta}) + \frac{\partial}{\partial \theta_k} \log p_{\mathcal{N}}(\mathbf{Y}^* \,|\, \mathbf{X}, \boldsymbol{\theta}) \right) + \sqrt{2\eta T}\, \nu_k, \tag{9}$$

To include automatic network rewiring in our deep learning model we adopt the approach described in Kappel et al. (2015). Instead of using the network parameters $\boldsymbol{\theta}$ directly to determine the synaptic weights of network $\mathcal{N}$, we apply a nonlinear transformation $w_k = f(\theta_k)$ to each connection $k$, given by the function

$$w_k \;=\; f(\theta_k) \;=\; s_k \frac{1}{\gamma} \log\left(1 + \exp(\gamma \, s_k \, \theta_k)\right) \;, \tag{10}$$

where $s_k \in \{1, -1\}$ is a parameter that determines the sign of the connection weight and $\gamma > 0$ is a constant parameter that determines the smoothness of the mapping. In the limit of large $\gamma$ Eq. (10) converges to the rectified linear function

$$w_k \;=\; f(\theta_k) \;=\; \begin{cases} 0 & \text{if } \theta_k < 0 \quad (\textit{dormant connection}) \\ s_k \, \theta_k & \text{else} \quad (\textit{active connection}) \end{cases} \;, \tag{11}$$

such that all connections with $\theta_k < 0$ are not functional.

Using this, the gradient of the log-likelihood function $\frac{\partial}{\partial \theta_k} \log p_{\mathcal{N}}(\mathbf{Y}^* \,|\, \mathbf{X}, \boldsymbol{\theta})$ in Eq. (9) can be written as $\frac{\partial}{\partial \theta_k} \log p_{\mathcal{N}}(\mathbf{Y}^* \,|\, \mathbf{X}, \boldsymbol{\theta}) = -\frac{\partial}{\partial \theta_k} f(\theta_k) \frac{\partial}{\partial \theta_k} E_{\mathbf{X}, \mathbf{Y}^*}(\boldsymbol{\theta})$ which for our choice of $f(\theta_k)$, Eqs. (10), becomes

$$\frac{\partial}{\partial \theta_k} \log p_{\mathcal{N}}(\mathbf{Y}^* \,|\, \mathbf{X}, \boldsymbol{\theta}) \;=\; -\sigma(\gamma \, s_k \, \theta_k) \, s_k \, \frac{\partial}{\partial \theta_k} E_{\mathbf{X}, \mathbf{Y}^*}(\boldsymbol{\theta}) \;, \tag{12}$$

where $\sigma(x) = \frac{1}{1+e^{-x}}$ denotes the sigmoid function. The error gradient $\frac{\partial}{\partial \theta_k} E_{\mathbf{X}, \mathbf{Y}^*}(\boldsymbol{\theta})$ can be computed using standard Error Backpropagation Neal (1992); Rumelhart et al. (1985).

Theorem 1 requires that Eq. (12) is twice differentiable, which is true for any finite value for $\gamma$. In our simulations we used the limiting case of large $\gamma$ such that dormant connections are actually mapped to zero weight. In this limit, one approaches the simple expression

$$\frac{\partial}{\partial \theta_k} \log p_{\mathcal{N}}(\mathbf{Y}^* \,|\, \mathbf{X}, \boldsymbol{\theta}) \;=\; \begin{cases} 0 & \text{if } \theta_k \le 0 \\ -s_k \frac{\partial}{\partial \theta_k} E_{\mathbf{X}, \mathbf{Y}^*}(\boldsymbol{\theta}) & \text{else} \end{cases} \;. \tag{13}$$

Thus, the gradient (13) vanishes for dormant connections ($\theta_k < 0$). Therefore changes of dormant connections are independent of the error gradient.

This leads to the parameter updates of the soft-DEEP R algorithm given by Eq. (8). The term $\sqrt{2T\eta}\, \nu_k$ results from the diffusion term $\mathcal{W}_k$ integrated over $\Delta t$, where $\nu_k$ is a Gaussian random variable with zero mean and unit variance. The term $-\eta\alpha$ results from the exponential prior distribution $p_{\mathcal{S}}(\boldsymbol{\theta})$ (the $\ell_1$-regularization). Note that this prior is not differentiable at 0. In (8) we approximate the gradient by assuming it to be zero at $\theta_k = 0$ and below. Thus, parameters on the negative axis are only driven by a random walk and parameter values might therefore diverge to $-\infty$. To fix this problem we introduced a reflecting boundary at $\theta_{\min}$ (parameters were clipped at this value). Another potential solution would be to use a different prior distribution that also effects the negative axis, however we found that Eq. (8) produces very good results in practice.

## D  ANALYSIS OF CONVERGENCE OF THE DEEP R ALGORITHM

Here we provide additional details to the convergence properties of the DEEP R algorithm. To do so we formulate the algorithm in terms of a Markov chain that evolves the parameters $\boldsymbol{\theta}$ and the connectivity constraints (listed in Algorithm 3). Each application of the Markov transition operators corresponds to one iteration of the DEEP R algorithm. We show that the distribution of parameters and network connectivities over the iterations of DEEP R converges to the stationary distribution Eq. (4) that jointly realizes parameter vectors $\boldsymbol{\theta}$ and admissible connectivity constraints.

Each iteration of DEEP R corresponds to two update steps, which we formally describe in Algorithm 3 using the Markov transition operators $\mathcal{T}_{\boldsymbol{\theta}}$ and $\mathcal{T}_{\mathbf{c}}$ and the binary constraint vector

1 **given:** initial values $\boldsymbol{\theta}'$, $\mathbf{c}'$ with $|\mathbf{c}'| = M$ ;
2 **for** $i$ $in$ $[1, N_{iterations}]$ **do**
3     $\boldsymbol{\theta} \sim \mathcal{T}_{\boldsymbol{\theta}}(\boldsymbol{\theta}|\boldsymbol{\theta}', \mathbf{c}')$ ;
4     $\mathbf{c} \sim \mathcal{T}_{\mathbf{c}}(\mathbf{c}|\boldsymbol{\theta})$ ;
5     $\boldsymbol{\theta}' \leftarrow \boldsymbol{\theta}, \mathbf{c}' \leftarrow \mathbf{c}$ ;
6 **end**

**Algorithm 3:** A reformulation of Algorithm 1 that is used for the proof in Theorem 2. Markov transition operators $\mathcal{T}_{\boldsymbol{\theta}}(\boldsymbol{\theta}|\boldsymbol{\theta}', \mathbf{c}')$ and $\mathcal{T}_{\mathbf{c}}(\mathbf{c}|\boldsymbol{\theta})$ are applied for parameter updates in each iteration. The transition operator $\mathcal{T}_{\boldsymbol{\theta}}(\boldsymbol{\theta}|\boldsymbol{\theta}', \mathbf{c}')$ updates $\boldsymbol{\theta}$ and corresponds to line 3, $\mathcal{T}_{\mathbf{c}}(\mathbf{c}|\boldsymbol{\theta})$ updates the connectivity constraint vector $\mathbf{c}$ and corresponds to lines 4,7 and 8 of Algorithm 1. $\boldsymbol{\theta}'$ and $\mathbf{c}'$ denote the parameter vector and connectivity constraint of the previous time step, respectively.

$\mathbf{c} \in \{0, 1\}^M$ over all $M$ connections of the network with elements $c_k$, where $c_k = 1$ represents an active connection $k$. $\mathbf{c}$ is a constraint on the dynamics, i.e., all connections $k$ for which $c_k = 0$ have to be dormant in the evolution of the parameters. The transition operators are conditional probability distributions from which in each iteration new samples for $\boldsymbol{\theta}$ and $\mathbf{c}$ are drawn for given previous values $\boldsymbol{\theta}'$ and $\mathbf{c}'$.

1. *Parameter update*: The transition operator $\mathcal{T}_{\boldsymbol{\theta}}(\boldsymbol{\theta}|\boldsymbol{\theta}', \mathbf{c}')$ updates all parameters $\theta_k$ for which $c_k = 1$ (active connections) and leaves the parameters $\theta_k$ at their current value for $c_k = 0$ (dormant connections). The update of active connections is realized by advancing the SDE (2) for an arbitrary time step $\Delta t$ (line 3 of Algorithm 3).

2. *Connectivity update*: for all parameters $\theta_k$ that are dormant, set $c_k = 0$ and randomly select an element $c_l$ which is currently 0 and set it to 1. This corresponds to line 3 of Algorithm 3 and is realized by drawing a new $\mathbf{c}$ from $\mathcal{T}_{\mathbf{c}}(\mathbf{c}|\boldsymbol{\theta})$.

The constraint imposed by $\mathbf{c}$ on $\boldsymbol{\theta}$ is formalized through the deterministic binary function $\mathcal{C}(\boldsymbol{\theta}, \mathbf{c}) \in \{0, 1\}$ which is 1 if the parameters $\boldsymbol{\theta}$ are compatible with the constraint vector $\mathbf{c}$ and 0 otherwise. This is expressed as (with $\Rightarrow$ denoting the Boolean implication):

$$\mathcal{C}(\boldsymbol{\theta}, \mathbf{c}) = \begin{cases} 1 & \text{if for all } k, 1 \leq k \leq K : c_k = 0 \Rightarrow \theta_k < 0 \\ 0 & \text{else} \end{cases} . \tag{14}$$

The constraint $\mathcal{C}(\boldsymbol{\theta}, \mathbf{c})$ is fulfilled if all connections $k$ with $c_k = 0$ are dormant ($\theta_k < 0$).

Note that the transition operator $\mathcal{T}_{\mathbf{c}}(\mathbf{c}|\boldsymbol{\theta})$ depends only on the parameter vector $\boldsymbol{\theta}$. It samples a new $\mathbf{c}$ with uniform probability among the constraint vectors that are compatible with the current set of parameters $\boldsymbol{\theta}$. We write the number of possible vectors $\mathbf{c}$ that are compatible with $\boldsymbol{\theta}$ as $\mu(\boldsymbol{\theta})$, given by the binomial coefficient (the number of possible selections that fulfill the constraint of new active connections)

$$\mu(\boldsymbol{\theta}) = \sum_{\mathbf{c} \in \chi} \mathcal{C}(\boldsymbol{\theta}, \mathbf{c}) = \binom{M - |\boldsymbol{\theta} \geq 0|}{K - |\boldsymbol{\theta} \geq 0|}, \quad \text{with} \quad \chi = \left\{ \boldsymbol{\xi} \in \{0, 1\}^M \mid |\boldsymbol{\xi}| = K \right\} , \tag{15}$$

where $|\mathbf{c}|$ denotes the number of non-zero elements in $\mathbf{c}$ and $\chi$ is the set of all binary vectors with exactly $K$ elements of value 1. Using this we can define the operator $\mathcal{T}_{\mathbf{c}}(\mathbf{c}|\boldsymbol{\theta})$ as:

$$\mathcal{T}_{\mathbf{c}}(\mathbf{c}|\boldsymbol{\theta}) = \frac{1}{\mu(\boldsymbol{\theta})} \sum_{\boldsymbol{\xi} \in \chi} \delta(\mathbf{c} - \boldsymbol{\xi}) \, \mathcal{C}(\boldsymbol{\theta}, \mathbf{c}) \tag{16}$$

where $\delta$ denotes the vectorized Kronecker delta function, with $\delta(\mathbf{0}) = 1$ and 0 else. Note that Eq. (16) assigns non-zero probability only to vectors $\mathbf{c}$ that are zero for elements $k$ for which $\theta_k < 0$ is true (assured by the second term). In addition vectors $\mathbf{c}$ have to fulfill $|\mathbf{c}| = K$. Therefore, sampling from this operator introduces randomly new connection for the number of missing ones in $\boldsymbol{\theta}$. This process models the *connectivity update* of Algorithm 3.

The transition operator $\mathcal{T}_{\boldsymbol{\theta}}(\boldsymbol{\theta}|\boldsymbol{\theta}', \mathbf{c}')$ in Eq. (34) evolves the parameter vector $\boldsymbol{\theta}$ under the constraint $\mathbf{c}$, i.e., it produces parameters confined to the connectivity constraint. By construction this operator has a stationary distribution that is given by the following Lemma.

**Lemma 1.** *Let $\mathcal{T}_{\boldsymbol{\theta}}(\boldsymbol{\theta}|\boldsymbol{\theta}', \mathbf{c})$ be the transition operator of the Markov chain over $\boldsymbol{\theta}$ which is defined, as the integration of the SDE written in Eq. (2) over an interval $\Delta t$ for active connections ($c_k = 1$), and as the identity for the remaining dormant connections ($c_k = 0$). Then it leaves the following distribution $p^*(\boldsymbol{\theta}|\mathbf{c})$ invariant*

$$p^*(\boldsymbol{\theta}|\mathbf{c}) = \frac{1}{p^*(\boldsymbol{\theta}_{\notin \mathbf{c}} < \mathbf{0})} p^*(\boldsymbol{\theta}) \mathcal{C}(\boldsymbol{\theta}, \mathbf{c}) \ , \tag{17}$$

*where $\boldsymbol{\theta}_{\in \mathbf{c}}$ denotes the truncation of the vector $\boldsymbol{\theta}$ to the active connections ($c_k = 1$), thus $p^*(\boldsymbol{\theta}_{\notin \mathbf{c}} < \mathbf{0})$ is the probability that all connections outside of $\mathbf{c}$ are dormant according to the posterior, and $p^*(\boldsymbol{\theta})$ is the posterior (see Theorem 1).*

The proof is divided into two sub proofs. First we show that the distribution defined as $p^*(\boldsymbol{\theta}|\mathbf{c}) = \frac{1}{\mathcal{L}(\mathbf{c})} p^*(\boldsymbol{\theta}) \mathcal{C}(\boldsymbol{\theta}, \mathbf{c})$ with $\mathcal{L}(\mathbf{c})$ a normalization constant, is left invariant by $\mathcal{T}_{\boldsymbol{\theta}}(\boldsymbol{\theta}|\boldsymbol{\theta}', \mathbf{c})$, second we will show that this normalization constant has to be equal to $p^*(\boldsymbol{\theta}_{\notin \mathbf{c}} < \mathbf{0})$. In coherence with the notation $\boldsymbol{\theta}_{\in \mathbf{c}}$ we will use verbally that $\theta_k$ is an element of $\mathbf{c}$ if $c_k = 1$.

*Proof.* To show that the distribution defined as $p^*(\boldsymbol{\theta}|\mathbf{c}) = \frac{1}{\mathcal{L}(\mathbf{c})} p^*(\theta) \mathcal{C}(\boldsymbol{\theta}, \mathbf{c})$ is left invariant, we will show directly that $\int_{\boldsymbol{\theta}'} \mathcal{T}_{\boldsymbol{\theta}}(\boldsymbol{\theta}|\boldsymbol{\theta}', \mathbf{c}) p^*(\boldsymbol{\theta}'|\mathbf{c}) d\boldsymbol{\theta}' = p^*(\boldsymbol{\theta}|\mathbf{c})$. To do so we will show that both $p^*(\boldsymbol{\theta}'|\mathbf{c})$ and $\mathcal{T}$ factorizes in terms that depend only on $\boldsymbol{\theta}'_{\in \mathbf{c}}$ or on $\boldsymbol{\theta}'_{\notin \mathbf{c}}$ and thus we will be able to separate the integral over $\boldsymbol{\theta}'$ as the product of two simpler integrals.

We first study the distribution $p^*(\boldsymbol{\theta}'_{\in \mathbf{c}}|\mathbf{c})$. Before factorizing, one has to notice a strong property of this distribution. Let's partition the tempered posterior distribution $p^*(\boldsymbol{\theta}')$ over the cases when the constraint is satisfied or not

$$p^*(\boldsymbol{\theta}'|\mathbf{c}) = \frac{1}{\mathcal{L}(\mathbf{c})} p^*(\boldsymbol{\theta}') \mathcal{C}(\mathbf{c}, \boldsymbol{\theta}) \tag{18}$$

$$= \frac{1}{\mathcal{L}(\mathbf{c})} \left[ p^*(\boldsymbol{\theta}', \mathcal{C}(\mathbf{c}, \boldsymbol{\theta}) = 1) + p^*(\boldsymbol{\theta}', \mathcal{C}(\mathbf{c}, \boldsymbol{\theta}) = 0) \right] \mathcal{C}(\mathbf{c}, \boldsymbol{\theta}) \tag{19}$$

when we multiply individually the first and the second term with $\mathcal{C}(\mathbf{c}, \boldsymbol{\theta})$, $\mathcal{C}(\mathbf{c}, \boldsymbol{\theta})$ can be replaced by its binary value and the second term is always null. It remains that

$$p^*(\boldsymbol{\theta}'|\mathbf{c}) = \frac{1}{\mathcal{L}(\mathbf{c})} p^*(\boldsymbol{\theta}', \mathcal{C}(\mathbf{c}, \boldsymbol{\theta}) = 1) \tag{20}$$

seeing that one can rewrite the condition $\mathcal{C}(\mathbf{c}, \boldsymbol{\theta}) = 1$ as the condition on the sign of the random variable $\boldsymbol{\theta}_{\notin \mathbf{c}} < \mathbf{0}$ (note that in this inequality $\mathbf{c}$ is a deterministic constant and $\boldsymbol{\theta}'_{\notin \mathbf{c}}$ is a random variable)

$$p^*(\boldsymbol{\theta}'|\mathbf{c}) = \frac{1}{\mathcal{L}(\mathbf{c})} p^*(\boldsymbol{\theta}', \boldsymbol{\theta}'_{\notin \mathbf{c}} < \mathbf{0}) \tag{21}$$

We can factorize the conditioned posterior as $p^*(\boldsymbol{\theta}, \boldsymbol{\theta}_{\notin \mathbf{c}} < \mathbf{0}) = p^*(\boldsymbol{\theta}_{\in \mathbf{c}}|\boldsymbol{\theta}_{\notin \mathbf{c}}, \boldsymbol{\theta}_{\notin \mathbf{c}} < \mathbf{0}) p^*(\boldsymbol{\theta}_{\notin \mathbf{c}}, \boldsymbol{\theta}_{\notin \mathbf{c}} < \mathbf{0})$. But when the dormant parameters are negative $\boldsymbol{\theta}_{\notin \mathbf{c}} < \mathbf{0}$, the active parameters $\boldsymbol{\theta}_{\in \mathbf{c}}$ do not depend on the actual value of the dormant parameters $\boldsymbol{\theta}_{\notin \mathbf{c}}$, so we can simplify the conditions of the first factor further to obtain

$$p^*(\boldsymbol{\theta}'|\mathbf{c}) = \frac{1}{\mathcal{L}(\mathbf{c})} p^*(\boldsymbol{\theta}'_{\in \mathbf{c}}|\boldsymbol{\theta}'_{\notin \mathbf{c}} < \mathbf{0}) p^*(\boldsymbol{\theta}'_{\notin \mathbf{c}}, \boldsymbol{\theta}'_{\notin \mathbf{c}} < \mathbf{0}) \tag{22}$$

We now study the operator $\mathcal{T}_{\boldsymbol{\theta}}$. It factorizes similarly because it is built out of two independent operations: one that integrates the SDE over the active connections and one that applies identity to the dormant ones. Moreover all the terms in the SDE which evolve the active parameters $\boldsymbol{\theta}_{\notin \mathbf{c}}$ are independent of the dormant ones $\boldsymbol{\theta}_{\notin \mathbf{c}}$ as long as we know they are dormant. Thus, the operator $\mathcal{T}_{\boldsymbol{\theta}}$ splits in two

$$\mathcal{T}_{\boldsymbol{\theta}}(\boldsymbol{\theta}|\boldsymbol{\theta}', \mathbf{c}) = \mathcal{T}_{\boldsymbol{\theta}}(\boldsymbol{\theta}_{\in \mathbf{c}}|\boldsymbol{\theta}'_{\in \mathbf{c}}, \mathbf{c}) \mathcal{T}_{\boldsymbol{\theta}}(\boldsymbol{\theta}_{\notin \mathbf{c}}|\boldsymbol{\theta}'_{\notin \mathbf{c}}, \mathbf{c}) \tag{23}$$

To finally separate the integration over $\boldsymbol{\theta}$ as a product of two integrals we need to make sure that all the factor depend only on the variable $\boldsymbol{\theta}'_{\in \mathbf{c}}$ or only on $\boldsymbol{\theta}'_{\notin \mathbf{c}}$. This might not seem obvious but even the conditioned probability $p^*(\boldsymbol{\theta}'_{\in \mathbf{c}}|\boldsymbol{\theta}'_{\notin \mathbf{c}} < \mathbf{0})$ is a function of $\boldsymbol{\theta}'_{\in \mathbf{c}}$ because in the conditioning

$\boldsymbol{\theta}'_{\notin\mathbf{c}} < \mathbf{0}$, $\boldsymbol{\theta}'_{\notin\mathbf{c}}$ refers to the random variable and not to a specific value over which we integrate. As a result the double integral is equal to the product of the two integrals

$$\int_{\boldsymbol{\theta}'} \mathcal{T}_{\boldsymbol{\theta}}(\boldsymbol{\theta}|\boldsymbol{\theta}', \mathbf{c})p^*(\boldsymbol{\theta}'|\mathbf{c})d\boldsymbol{\theta}' = \frac{1}{\mathcal{L}(\mathbf{c})}\int_{\boldsymbol{\theta}'_{\in\mathbf{c}}} \mathcal{T}_{\boldsymbol{\theta}}(\boldsymbol{\theta}_{\in\mathbf{c}}|\boldsymbol{\theta}'_{\in\mathbf{c}}, \mathbf{c})p^*(\boldsymbol{\theta}'_{\in\mathbf{c}}|\boldsymbol{\theta}'_{\notin\mathbf{c}} < \mathbf{0})d\boldsymbol{\theta}'_{\in\mathbf{c}} \quad (24)$$

$$\int_{\boldsymbol{\theta}'_{\notin\mathbf{c}}} \mathcal{T}_{\boldsymbol{\theta}}(\boldsymbol{\theta}_{\notin\mathbf{c}}|\boldsymbol{\theta}'_{\notin\mathbf{c}}, \mathbf{c})p^*(\boldsymbol{\theta}'_{\notin\mathbf{c}}, \boldsymbol{\theta}'_{\notin\mathbf{c}} < \mathbf{0})d\boldsymbol{\theta}'_{\notin\mathbf{c}} \quad (25)$$

We can now study the two integrals separately. The second integral over the parameters $\boldsymbol{\theta}_{\notin\mathbf{c}}$ is simpler because by construction the operator $\mathcal{T}_{\boldsymbol{\theta}}$ is the identity

$$\int_{\boldsymbol{\theta}'_{\notin\mathbf{c}}} \mathcal{T}_{\boldsymbol{\theta}}(\boldsymbol{\theta}_{\notin\mathbf{c}}|\boldsymbol{\theta}'_{\notin\mathbf{c}}, \mathbf{c})p^*(\boldsymbol{\theta}'_{\notin\mathbf{c}}, \boldsymbol{\theta}'_{\notin\mathbf{c}} < \mathbf{0})d\boldsymbol{\theta}'_{\notin\mathbf{c}} = p^*(\boldsymbol{\theta}_{\notin\mathbf{c}}, \boldsymbol{\theta}_{\notin\mathbf{c}} < \mathbf{0}) \quad (26)$$

There is more to say about the first integral over the active connections $\boldsymbol{\theta}_{\in\mathbf{c}}$. The operator $\mathcal{T}_{\boldsymbol{\theta}}(\boldsymbol{\theta}_{\in\mathbf{c}}|\boldsymbol{\theta}'_{\notin\mathbf{c}}, \mathbf{c})$ integrates over the active parameters $\boldsymbol{\theta}_{\in\mathbf{c}}$ the same SDE as before with the difference that the network is reduced to a sparse architecture where only the parameters $\boldsymbol{\theta}_{\in\mathbf{c}}$ are active. We want to find the relationship between the stationary distribution of this new operator and $p^*(\boldsymbol{\theta})$ that is written in the integral which is defined in equation (3) as the tempered posterior of the dense network. In fact, the tempered posterior of the dense network marginalized and conditioned over the dormant connections $p^*(\boldsymbol{\theta}'_{\in\mathbf{c}}|\boldsymbol{\theta}'_{\notin\mathbf{c}} < \mathbf{0})$ is equal to the stationary distribution of $\mathcal{T}_{\boldsymbol{\theta}}(\boldsymbol{\theta}_{\in\mathbf{c}}|\boldsymbol{\theta}'_{\notin\mathbf{c}}, \mathbf{c})$ (i.e. of the SDE in the sparse network). To prove this, we detail in the following paragraph that the drift in the SDE evolving the sparse network is given by the log-posterior of the dense network condition on $\boldsymbol{\theta}_{\notin\mathbf{c}} < \mathbf{0}$ and using Theorem 1, we will conclude that $p^*(\boldsymbol{\theta}'_{\in\mathbf{c}}|\boldsymbol{\theta}'_{\notin\mathbf{c}} < \mathbf{0})$ is the stationary distribution of $\mathcal{T}_{\boldsymbol{\theta}}(\boldsymbol{\theta}_{\in\mathbf{c}}|\boldsymbol{\theta}'_{\notin\mathbf{c}}, \mathbf{c})$.

We write the prior and the likelihood of the sparse network as function of the prior and the likelihood $p_{\mathcal{S}}$ with $p_{\mathcal{N}}$ of the dense network. The likelihood in the sparse network is defined as previously with the exception that the dormant connections are given zero-weight $w_k = 0$ so it is equal to $p_{\mathcal{N}}(\mathbf{X}, \mathbf{Y}^*|\boldsymbol{\theta}_{\in\mathbf{c}}, \boldsymbol{\theta}_{\notin\mathbf{c}} < \mathbf{0})$. The difference between the prior that defines soft-DEEP R and the prior of DEEP R remains in the presence of the constraint. When considering the sparse network defined by $\mathbf{c}$ the constraint is satisfied and the prior of soft-DEEP R marginalized over the dormant connections $p_{\mathcal{S}}(\boldsymbol{\theta}_{\in\mathbf{c}})$ is the prior of the sparse network with $p_{\mathcal{S}}$ defined as before. As this prior is connection-specific ($p_{\mathcal{S}}(\theta_i)$ independent of $\theta_j$), this implies that $p_{\mathcal{S}}(\boldsymbol{\theta}_{\in\mathbf{c}})$ is independent of the dormant connection, and the prior $p_{\mathcal{S}}(\boldsymbol{\theta}_{\in\mathbf{c}})$ is equal to $p_{\mathcal{S}}(\boldsymbol{\theta}_{\in\mathbf{c}}|\boldsymbol{\theta}_{\notin\mathbf{c}} < \mathbf{0})$. Thus, we can write the posterior of the sparse network which is by definition proportional to the product $p_{\mathcal{N}}(\mathbf{X}, \mathbf{Y}^*|\boldsymbol{\theta}_{\in\mathbf{c}}, \boldsymbol{\theta}_{\notin\mathbf{c}} < \mathbf{0})p_{\mathcal{S}}(\boldsymbol{\theta}_{\in\mathbf{c}}|\boldsymbol{\theta}_{\notin\mathbf{c}} < \mathbf{0})$. Looking back to the definition of the posterior of the dense network this product is actually proportional to posterior of the dense network conditioned on the negativity of dormant connections $p^*(\boldsymbol{\theta}_{\in\mathbf{c}}|\boldsymbol{\theta}_{\notin\mathbf{c}} < \mathbf{0}, \mathbf{X}, \mathbf{Y}^*)$. The posterior of the sparse network is therefore proportional to the conditioned posterior of the dense network but as they both normalize to 1 they are actually equal. Writing down the new SDE, the diffusion term $\sqrt{2T\beta}d\mathcal{W}_k$ remains unchanged, and the drift term is given by the gradient of the log-posterior $\log p^*(\boldsymbol{\theta}_{\in\mathbf{c}}|\boldsymbol{\theta}_{\notin\mathbf{c}} < \mathbf{0}, \mathbf{X}, \mathbf{Y}^*)$. Applying Theorem 1 to this new SDE, we now confirm that the tempered and conditioned posterior of the dense network $p^*(\boldsymbol{\theta}_{\in\mathbf{c}}|\boldsymbol{\theta}_{\notin\mathbf{c}} < \mathbf{0})$ is left invariant by the SDE evolving the sparse network. As $\mathcal{T}_{\boldsymbol{\theta}}(\boldsymbol{\theta}_{\in\mathbf{c}}|\boldsymbol{\theta}'_{\notin\mathbf{c}}, \mathbf{c})$ is the integration for a given $\Delta t$ of this SDE, it also leaves $p^*(\boldsymbol{\theta}_{\in\mathbf{c}}|\boldsymbol{\theta}_{\notin\mathbf{c}} < \mathbf{0})$ invariant. This yields

$$\int_{\boldsymbol{\theta}'_{\in\mathbf{c}}} \mathcal{T}_{\boldsymbol{\theta}}(\boldsymbol{\theta}_{\in\mathbf{c}}|\boldsymbol{\theta}'_{\in\mathbf{c}}, \mathbf{c})p^*(\boldsymbol{\theta}'_{\in\mathbf{c}}|\boldsymbol{\theta}'_{\notin\mathbf{c}} < \mathbf{0})d\boldsymbol{\theta}'_{\in\mathbf{c}} = p^*(\boldsymbol{\theta}_{\in\mathbf{c}}|\boldsymbol{\theta}_{\notin\mathbf{c}} < \mathbf{0}) \quad (27)$$

As we simplified both integrals we arrived at

$$\int_{\boldsymbol{\theta}'} \mathcal{T}_{\boldsymbol{\theta}}(\boldsymbol{\theta}|\boldsymbol{\theta}', \mathbf{c})p^*(\boldsymbol{\theta}'|\mathbf{c})d\boldsymbol{\theta}' = \frac{1}{\mathcal{L}(\mathbf{c})}p^*(\boldsymbol{\theta}_{\in\mathbf{c}}|\boldsymbol{\theta}_{\notin\mathbf{c}} < \mathbf{0})p^*(\boldsymbol{\theta}_{\notin\mathbf{c}}, \boldsymbol{\theta}_{\notin\mathbf{c}} < \mathbf{0}) \quad (28)$$

Replacing the right-end side with equation (22) we conclude

$$\int_{\boldsymbol{\theta}'} \mathcal{T}_{\boldsymbol{\theta}}(\boldsymbol{\theta}|\boldsymbol{\theta}', \mathbf{c})p^*(\boldsymbol{\theta}'|\mathbf{c})d\boldsymbol{\theta}' = p^*(\boldsymbol{\theta}|\mathbf{c}) \quad (29)$$

$\square$

We now show that the normalization constant $\mathcal{L}(\mathbf{c})$ is equal to $p^*(\boldsymbol{\theta}_{\notin \mathbf{c}} < \mathbf{0})$.

*Proof.* Using equation (22), as $p^*(\theta|\mathbf{c})$ normalizes to 1 the normalization constant is equal to

$$\mathcal{L}(\mathbf{c}) \;\; = \;\; \int_{\boldsymbol{\theta}} p^*(\boldsymbol{\theta}_{\in \mathbf{c}}|\boldsymbol{\theta}_{\notin \mathbf{c}} < \mathbf{0}) p^*(\boldsymbol{\theta}_{\notin \mathbf{c}}, \boldsymbol{\theta}_{\notin \mathbf{c}} < \mathbf{0}) d\boldsymbol{\theta} \tag{30}$$

By factorizing the last factor in the integral we have that

$$p^*(\boldsymbol{\theta}, \boldsymbol{\theta}_{\notin \mathbf{c}} < \mathbf{0}) \;\; = \;\; p^*(\boldsymbol{\theta}_{\in \mathbf{c}}|\boldsymbol{\theta}_{\notin \mathbf{c}} < \mathbf{0}) p^*(\boldsymbol{\theta}_{\notin \mathbf{c}}|\boldsymbol{\theta}_{\notin \mathbf{c}} < \mathbf{0}) p^*(\boldsymbol{\theta}_{\notin \mathbf{c}} < \mathbf{0}) \tag{31}$$

The last term does not depend on the value $\boldsymbol{\theta}$ because $\boldsymbol{\theta}_{\notin \mathbf{c}}$ refers here to the random variable and the first two term depend either on $\boldsymbol{\theta}_{\in \mathbf{c}}$ or $\boldsymbol{\theta}_{\notin \mathbf{c}}$. Plugging the previous equation into the computation of $\mathcal{L}(\mathbf{c})$ and separating the integrals we have

$$\mathcal{L}(\mathbf{c}) \;\; = \;\; p^*(\boldsymbol{\theta}_{\notin \mathbf{c}} < \mathbf{0}) \underbrace{\int_{\boldsymbol{\theta}_{\in \mathbf{c}}} p^*(\boldsymbol{\theta}_{\in \mathbf{c}}|\boldsymbol{\theta}_{\notin \mathbf{c}} < \mathbf{0}) d\boldsymbol{\theta}_{\in \mathbf{c}}}_{=1} \underbrace{\int_{\boldsymbol{\theta}_{\notin \mathbf{c}}} p^*(\boldsymbol{\theta}_{\notin \mathbf{c}}|\boldsymbol{\theta}_{\notin \mathbf{c}} < \mathbf{0}) d\boldsymbol{\theta}_{\notin \mathbf{c}}}_{=1} \tag{32}$$

$\square$

Due to Lemma 1, there exists a distribution $\pi(\boldsymbol{\theta} \,|\, \mathbf{c})$ of the following form which is left invariant by the operator $\mathcal{T}_{\boldsymbol{\theta}}$

$$\pi(\boldsymbol{\theta} \,|\, \mathbf{c}) \;\; = \;\; \frac{1}{\mathcal{L}(\mathbf{c})} \pi(\boldsymbol{\theta}) \, \mathcal{C}(\boldsymbol{\theta}, \mathbf{c}) \,, \tag{33}$$

where $\mathcal{L}(\mathbf{c})$ is a normalizer and where $\pi(\boldsymbol{\theta})$ is some distribution over $\boldsymbol{\theta}$ that may not obey the constraint $\mathcal{C}(\boldsymbol{\theta}, \mathbf{c})$. This will imply a very strong property on the compound operator which evolves both $\boldsymbol{\theta}$ and $\mathbf{c}$. To form $\mathcal{T}$ the operators $\mathcal{T}_{\boldsymbol{\theta}}$ and $\mathcal{T}_{\mathbf{c}}$ are performed one after the other so that the total update can be written in terms of the compound operator

$$\mathcal{T}(\boldsymbol{\theta}, \mathbf{c}|\boldsymbol{\theta}', \mathbf{c}') = \mathcal{T}_{\mathbf{c}}(\mathbf{c}|\boldsymbol{\theta}) \mathcal{T}_{\boldsymbol{\theta}}(\boldsymbol{\theta}|\boldsymbol{\theta}', \mathbf{c}') \,. \tag{34}$$

Applying the compound operator $\mathcal{T}$ given by Eq. (34) corresponds to advancing the parameters for a single iteration of Algorithm 3.

Using these definitions a general theorem can be enunciated for arbitrary distributions $\pi(\boldsymbol{\theta} \,|\, \mathbf{c})$ of the form (33). The following theorem states that the distribution of variable pairs $\mathbf{c}$ and $\boldsymbol{\theta}$ that is left stationary by the operator $\mathcal{T}$ is the product of Eq. (33) and a uniform prior $p_{\mathcal{C}}(\mathbf{c})$ over the constraint vectors which have $K$ active connections. This prior is formally defined as

$$p_{\mathcal{C}}(\mathbf{c}) \;\; = \;\; \frac{1}{|\mathcal{X}|} \sum_{\boldsymbol{\xi} \in \chi} \delta(\mathbf{c} - \boldsymbol{\xi}) \,, \tag{35}$$

with $\chi$ as defined in (15). The theorem to analyze the dynamics of Algorithm 3 can now be written as

**Theorem 2.** *Let $\mathcal{T}_{\boldsymbol{\theta}}(\boldsymbol{\theta}|\boldsymbol{\theta}', \mathbf{c})$ be the transition operator of a Markov chain over $\boldsymbol{\theta}$ and let $\mathcal{T}_{\mathbf{c}}(\mathbf{c}|\boldsymbol{\theta})$ be defined by Eq. (16). Under the assumption that $\mathcal{T}_{\boldsymbol{\theta}}(\boldsymbol{\theta}|\boldsymbol{\theta}', \mathbf{c})$ has a unique stationary distribution $\pi(\boldsymbol{\theta}|\mathbf{c})$, that verifies Eq. (33), then the Markov chain over $\boldsymbol{\theta}$ and $\mathbf{c}$ with transition operator*

$$\mathcal{T}(\boldsymbol{\theta}, \mathbf{c}|\boldsymbol{\theta}', \mathbf{c}') = \mathcal{T}_{\mathbf{c}}(\mathbf{c}|\boldsymbol{\theta}) \mathcal{T}_{\boldsymbol{\theta}}(\boldsymbol{\theta}|\boldsymbol{\theta}', \mathbf{c}') \tag{36}$$

*leaves the stationary distribution*

$$p^*(\boldsymbol{\theta}, \mathbf{c}) = \frac{\mathcal{L}(\mathbf{c})}{\sum_{\mathbf{c}' \in \mathcal{X}} \mathcal{L}(\mathbf{c}')} \pi(\boldsymbol{\theta}|\mathbf{c}) p_{\mathcal{C}}(\mathbf{c}) \tag{37}$$

*invariant. If the Markov chain of the transition operator $\mathcal{T}_{\boldsymbol{\theta}}(\boldsymbol{\theta}|\boldsymbol{\theta}', \mathbf{c}')$ is ergodic, then the stationary distribution is also unique.*

*Proof.* Theorem 2 holds for $\mathcal{T}_{\mathbf{c}}$ in combination with any operator $\mathcal{T}_{\boldsymbol{\theta}}$ that updates $\boldsymbol{\theta}$ that can be written in the form (33). We prove Theorem 2 by proving the following equality to show that $\mathcal{T}$ leaves (37) invariant:

$$\sum_{\mathbf{c}'} \int \mathcal{T}(\boldsymbol{\theta}, \mathbf{c}|\boldsymbol{\theta}', \mathbf{c}') \, p^*(\boldsymbol{\theta}', \mathbf{c}') \, \mathrm{d}\boldsymbol{\theta}' \;\; = \;\; p^*(\boldsymbol{\theta}, \mathbf{c}) \,. \tag{38}$$

We expand the left-hand term using Eq. (36) and Eq. (37)

$$\sum_{\mathbf{c}'} \int \mathcal{T}(\boldsymbol{\theta}, \mathbf{c}|\boldsymbol{\theta}', \mathbf{c}') \, p^*(\boldsymbol{\theta}', \mathbf{c}') \, \mathrm{d}\boldsymbol{\theta}' \; =$$

$$\sum_{\mathbf{c}'} \int \mathcal{T}_{\mathbf{c}}(\mathbf{c}|\boldsymbol{\theta}) \mathcal{T}_{\boldsymbol{\theta}}(\boldsymbol{\theta}|\boldsymbol{\theta}', \mathbf{c}') \frac{\mathcal{L}(\mathbf{c}')}{\sum_{\mathbf{c}'' \in \mathcal{X}} \mathcal{L}(\mathbf{c}'')} \pi(\boldsymbol{\theta}'|\mathbf{c}') p_{\mathcal{C}}(\mathbf{c}') \, \mathrm{d}\boldsymbol{\theta}' \; . \qquad (39)$$

Since $\mathcal{T}_{\mathbf{c}}$ does not depend on $\boldsymbol{\theta}'$ and $\mathbf{c}'$, one can pull it out of the sum and integral and then marginalize over $\boldsymbol{\theta}'$ by observing that $\pi(\boldsymbol{\theta}|\mathbf{c}')$ is by definition the stationary distribution of $\mathcal{T}_{\boldsymbol{\theta}}(\boldsymbol{\theta}|\boldsymbol{\theta}', \mathbf{c}')$:

$$\sum_{\mathbf{c}'} \int \mathcal{T}(\boldsymbol{\theta}, \mathbf{c}|\boldsymbol{\theta}', \mathbf{c}') \, p^*(\boldsymbol{\theta}', \mathbf{c}') \, \mathrm{d}\boldsymbol{\theta}' \; = \qquad (40)$$

$$= \; \mathcal{T}_{\mathbf{c}}(\mathbf{c}|\boldsymbol{\theta}) \sum_{\mathbf{c}'} \int \mathcal{T}_{\boldsymbol{\theta}}(\boldsymbol{\theta}|\boldsymbol{\theta}', \mathbf{c}') \frac{\mathcal{L}(\mathbf{c}')}{\sum_{\mathbf{c}'' \in \mathcal{X}} \mathcal{L}(\mathbf{c}'')} \pi(\boldsymbol{\theta}'|\mathbf{c}') p_{\mathcal{C}}(\mathbf{c}') \, \mathrm{d}\boldsymbol{\theta}' \qquad (41)$$

$$= \; \mathcal{T}_{\mathbf{c}}(\mathbf{c}|\boldsymbol{\theta}) \sum_{\mathbf{c}'} \frac{\mathcal{L}(\mathbf{c}')}{\sum_{\mathbf{c}'' \in \mathcal{X}} \mathcal{L}(\mathbf{c}'')} \pi(\boldsymbol{\theta}|\mathbf{c}') p_{\mathcal{C}}(\mathbf{c}') \; . \qquad (42)$$

What remains to be done is to marginalize over $\mathbf{c}'$ and to relate the result to the stationary distribution $p^*(\boldsymbol{\theta}, \mathbf{c}) = \frac{\mathcal{L}(\mathbf{c})}{\sum_{\mathbf{c}' \in \mathcal{X}} \mathcal{L}(\mathbf{c}')} \pi(\boldsymbol{\theta}|\mathbf{c}) p_{\mathcal{C}}(\mathbf{c})$. First we replace $\mathcal{T}_{\mathbf{c}}$ with its definition Eq. (16):

$$\sum_{\mathbf{c}'} \int \mathcal{T}(\boldsymbol{\theta}, \mathbf{c}|\boldsymbol{\theta}', \mathbf{c}') \, p^*(\boldsymbol{\theta}', \mathbf{c}') \, \mathrm{d}\boldsymbol{\theta}' \; =$$

$$= \; \left( \frac{1}{\mu(\boldsymbol{\theta})} \sum_{\boldsymbol{\xi} \in \boldsymbol{\chi}} \delta(\mathbf{c} - \boldsymbol{\xi}) \right) \mathcal{C}(\boldsymbol{\theta}, \mathbf{c}) \sum_{\mathbf{c}'} \frac{\mathcal{L}(\mathbf{c}')}{\sum_{\mathbf{c}'' \in \mathcal{X}} \mathcal{L}(\mathbf{c}'')} \pi(\boldsymbol{\theta}|\mathbf{c}') p_{\mathcal{C}}(\mathbf{c}')$$

As the operator $\mathcal{T}_{\mathbf{c}}$ samples uniform across admissible configurations it has a close relationship with the uniform probability distribution $p_{\mathcal{C}}$ and we can now replace the sum over $\boldsymbol{\xi}$ using Eq. (35)

$$\sum_{\mathbf{c}'} \int \mathcal{T}(\boldsymbol{\theta}, \mathbf{c}|\boldsymbol{\theta}', \mathbf{c}') \, p^*(\boldsymbol{\theta}', \mathbf{c}') \, \mathrm{d}\boldsymbol{\theta}' \; = \; \frac{|\boldsymbol{\chi}|}{\mu(\boldsymbol{\theta})} p_{\mathcal{C}}(\mathbf{c}) \mathcal{C}(\boldsymbol{\theta}, \mathbf{c}) \sum_{\mathbf{c}'} \frac{\mathcal{L}(\mathbf{c}')}{\sum_{\mathbf{c}'' \in \mathcal{X}} \mathcal{L}(\mathbf{c}'')} \pi(\boldsymbol{\theta}|\mathbf{c}') p_{\mathcal{C}}(\mathbf{c}') \; .$$

From Eq. (15), Eq. (33) and Eq. (35) we find the equalities $\sum_{\mathbf{c}'} \mathcal{L}(\mathbf{c}') \pi(\boldsymbol{\theta}|\mathbf{c}') p_{\mathcal{C}}(\mathbf{c}') = \pi(\boldsymbol{\theta}) \sum_{\mathbf{c}'} \mathcal{C}(\boldsymbol{\theta}, \mathbf{c}') p_{\mathcal{C}}(\mathbf{c}') = \frac{\mu(\boldsymbol{\theta})}{|\boldsymbol{\chi}|} \pi(\boldsymbol{\theta})$. Using this we get

$$\sum_{\mathbf{c}'} \int \mathcal{T}(\boldsymbol{\theta}, \mathbf{c}|\boldsymbol{\theta}', \mathbf{c}') \, p^*(\boldsymbol{\theta}', \mathbf{c}') \, \mathrm{d}\boldsymbol{\theta}' \; = \; \frac{|\boldsymbol{\chi}|}{\mu(\boldsymbol{\theta})} p_{\mathcal{C}}(\mathbf{c}) \mathcal{C}(\boldsymbol{\theta}, \mathbf{c}) \frac{1}{\sum_{\mathbf{c}' \in \mathcal{X}} \mathcal{L}(\mathbf{c}')} \frac{\mu(\boldsymbol{\theta})}{|\boldsymbol{\chi}|} \pi(\boldsymbol{\theta})$$

Finally using again Eq. (33), i.e. $\pi(\boldsymbol{\theta}) \mathcal{C}(\boldsymbol{\theta}, \mathbf{c}) = \mathcal{L}(\mathbf{c}) \pi(\boldsymbol{\theta}|\mathbf{c})$

$$\sum_{\mathbf{c}'} \int \mathcal{T}(\boldsymbol{\theta}, \mathbf{c}|\boldsymbol{\theta}', \mathbf{c}') \, p^*(\boldsymbol{\theta}', \mathbf{c}') \, \mathrm{d}\boldsymbol{\theta}' \; = \; \frac{\mathcal{L}(\mathbf{c})}{\sum_{\mathbf{c}' \in \mathcal{X}} \mathcal{L}(\mathbf{c}')} \pi(\boldsymbol{\theta}|\mathbf{c}) \, p_{\mathcal{C}}(\mathbf{c}) \; = \; p^*(\boldsymbol{\theta}, \mathbf{c}) \; .$$

This shows that the stationary distribution Eq. (37) is invariant under the compound operator (36). Under the assumption that $\mathcal{T}_{\boldsymbol{\theta}}(\boldsymbol{\theta}|\boldsymbol{\theta}', \mathbf{c}')$ is ergodic it allows each parameter $\theta_k$ to become negative with non-zero probability and the stationary distribution is also unique. This can be seen by noting that under this assumption each connection will become dormant sooner or later and thus each state in $\mathbf{c}$ can be reached from any other state $\mathbf{c}'$. The Markov chain is therefore irreducible and the stationary distribution is unique. $\qquad \square$

Lemma 1 provides for the case of algorithm 3 the existence of an invariant distribution that is needed to apply Theorem 2. We conclude that the distribution $p^*(\boldsymbol{\theta}, \mathbf{c})$ defined by plugging the result of Lemma 1 Eq. (17) into the result of Theorem 2 Eq. (37), is left invariant by algorithm 3 and it is written

$$\boxed{p^*(\boldsymbol{\theta}, \mathbf{c}) = \frac{p^*(\boldsymbol{\theta}_{\notin \mathbf{c}} < \mathbf{0})}{\sum_{\mathbf{c}' \in \mathcal{X}} p^*(\boldsymbol{\theta}_{\notin \mathbf{c}'} < \mathbf{0})} p^*(\boldsymbol{\theta}|\mathbf{c}) p_{\mathcal{C}}(\mathbf{c})} \qquad (43)$$

Where $p^*(\boldsymbol{\theta})$ is defined as previously as the tempered posterior of the dense network which is left invariant by soft-DEEP R according to Theorem 1. The prior $p_\mathcal{C}(\mathbf{c})$ in Eq. (43) assures that only constraints $\mathbf{c}$ with exactly $K$ active connections are selected. By Theorem 2 the stationary distribution (43) is also unique. By inserting the result of Lemma 1, Eq. (17) we recover Eq. 4 of the main text.

Interestingly, by marginalizing over $\boldsymbol{\theta}$, we can show that the network architecture identified by $\mathbf{c}$ is sampled by algorithm 3 from the probability distribution

$$p^*(\mathbf{c}) = \frac{p^*(\boldsymbol{\theta}_{\notin\mathbf{c}} < \mathbf{0})}{\sum_{\mathbf{c}' \in \mathcal{X}} p^*(\boldsymbol{\theta}_{\notin\mathbf{c}'} < \mathbf{0})} p_\mathcal{C}(\mathbf{c}) \tag{44}$$

The difference between the formal algorithm 3 and the actual implementation of DEEP R are that $\mathcal{T}_\mathbf{c}$ keeps the dormant connection parameters constant, whereas in DEEP R we implement this by setting connections to 0 as they become negative. We found the process used in DEEP R works very well in practice. The reason why we did not implement algorithm 3 in practice is that we did not want to consume memory by storing any parameter for dormant connections. This difference is obsolete from the view point of the network function for a given $(\boldsymbol{\theta}, \mathbf{c})$ pair because nor negative neither strictly zero $\theta$ have any influence on the network function.

This difference might seem problematic to consider that the properties of convergence to a specific stationary distribution as proven for algorithm 3 extends to DEEP R. However, both the theorem and the implementation are rather unspecific regarding the choice of the prior on the negative sides $\theta < 0$. We believe that, with good choices of priors on the negative side, the conceptual and quantitative difference between the distribution explored by 3 and DEEP R are minor, and in general, algorithm 3 is a decent mathematical formalization of DEEP R for the purpose of this paper.

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
