# OpenReview forum: "Deep Rewiring: Training very sparse deep networks"
_ICLR.cc/2018/Conference — Accept (Poster)_

### Official Review · AnonReviewer3 · 2017-11-27
**Review of "DEEP R"**

**Rating:** 8
**Confidence:** 4

**Review:**

In this paper, the authors present an approach to implement deep learning directly on sparsely connected graphs. Previous approaches have focused on transferring trained deep networks to a sparse graph for fast or efficient utilization; using this approach, sparse networks can be trained efficiently online, allowing for fast and flexible learning. Further investigation is necessary to understand the full implications of the two main conceptual changes introduced here (signed connections that can disappear and random walk in parameter space), but the initial results are quite promising.

It would also be interesting to understand more fully how performance scales to larger networks. If the target connectivity could be pushed to a very sparse limit, where only a fixed number of connections were added with each additional neuron, then this could significantly shape how these networks are trained at very large scales. Perhaps the heuristics for initializing the connectivity matrices will be insufficient, but could these be improved in further work?

As a last minor comment, the authors should specify explicitly what the shaded areas are in Fig. 4b,c.

---

### Official Review · AnonReviewer1 · 2017-11-30
**Interesting algorithm to training with limited memory, but needs some additional relationships to existing work.**

**Rating:** 5
**Confidence:** 5

**Review:**

The authors provide a novel, interesting, and simple algorithm capable of training with limited memory.  The algorithm is well-motivated and clearly explained, and empirical evidence suggests that the algorithm works well.  However, the paper needs additional examination in how the algorithm can deal with larger data inputs and outputs.  Second, the relationship to existing work needs to be explained better.

Pro:
The algorithm is clearly explained, well-motivated, and empirically supported.

Con:
The relationship to stochastic gradient markov chain monte carlo needs to be explained better.  In particular, the update form was first introduced in [1], the annealing scheme was analyzed in [2], and the reflection step was introduced in [3].  These relationships need to be explained clearly.
The evidence is presented on very small input data.  With something like natural images, the parameterization is much larger and with more data, the number of total parameters is much larger.  Is there any evidence that the proposed algorithm could continue performing comparatively as the total number of parameters in state-of-the-art networks increases? This would require a smaller ratio of included parameters.

[1] Welling, M. and Teh, Y.W., 2011. Bayesian learning via stochastic gradient Langevin dynamics. In Proceedings of the 28th International Conference on Machine Learning (ICML-11)(pp. 681-688).

[2] Chen, C., Carlson, D., Gan, Z., Li, C. and Carin, L., 2016, May. Bridging the gap between stochastic gradient MCMC and stochastic optimization. In Artificial Intelligence and Statistics(pp. 1051-1060).

[3] Patterson, S. and Teh, Y.W., 2013. Stochastic gradient Riemannian Langevin dynamics on the probability simplex. In Advances in Neural Information Processing Systems (pp. 3102-3110).

---

### Official Review · AnonReviewer2 · 2017-12-02
**Promising approach for network compression on hardware with limited resources, however important references to previous work are missing**

**Rating:** 6
**Confidence:** 4

**Review:**

This paper presents an iterative approach to sparsify a network already during training. During the training process, the amount of connections in the network is guaranteed to stay under a specific threshold. This is a big advantage when training is performed on hardware with computational limitations, in comparison to "post-hoc" sparsification methods, that compress the network after training.
The method is derived by considering the "rewiring" of an (artificial) neural network as a stochastic process. This perspective is based on a recent model in computational biology but also can be interpreted as a (sequential) monte carlo sampling based stochastic gradient descent approach. References to previous work in this area are missing, e.g.

[1] de Freitas et al., Sequential Monte Carlo Methods to Train Neural Network
Models, Neural Computation 2000
[2] Welling et al., Bayesian Learning via Stochastic Gradient Langevin Dynamics, ICML 2011

Especially the stochastic gradient method in [2] is strongly related to the existing approach.

Positive aspects

- The presented approach is well grounded in the theory of stochastic processes. The authors provide proofs of convergence by showing that the iterative updates converge to a fixpoint of the stochastic process

- By keeping the temperature parameter of the stochastic process high, it can be directly applied to online transfer learning.

- The method is specifically designed for online learning with limited hardware ressources.

Negative aspects

- The presented approach is outperformed for moderate compression levels (by Han's pruning method for >5% connectivity on MNIST, Fig. 3 A, and by l1-shrinkage for >40% connectivity on CIFAR-10 and TIMIT, Fig. 3 B&C). Especially the results on MNIST suggest that this method is most advantageous for very high compression levels. However in these cases the overall classification accuracy has already dropped significantly which could limit the practical applicability.

- A detailled discussion of the relation to previously existing very similar work is missing (see above)


Technical Remarks

Fig. 1, 2 and 3 are referenced on the pages following the page containing the figure. Readibility could be slightly increased by putting the figures on the respective pages.

---

### Author Response · Authors · 2018-08-07
**Erratum: Corrections in the appendix**

In the section "Methods" and sub-section "Initialization of connectivity matrices" of the appendix:

The fifth paragraph should read:
"For CIFAR-10 [...] The numbers of parameters per connectivity matrices were therefore 5k, 102k, 885k, 74k and 2k from input to output. The connectivity matrices were initialized with connectivity $1, 4p_0, 0.4p_0, 4p_0,$ and $1$."
(before the number of parameters in the fourth connection matrix was mistaken as 738K and the connectivities  were $1, 8p_0, 0.8p_0, 8p_0,$ and $1$)

The sixth paragraph should read:
"- For TIMIT, [..] Each of these three connectivity matrices were initialized with a connectivity of $1.8 p_0, 0.6 p_0$, and $ 6 p_0$."
(before these numbers were mistaken as $3 p_0, p_0$, and $ 10 p_0$)

---

### Decision · Program_Chairs · 2018-01-29
**ICLR 2018 Conference Acceptance Decision**

**Decision:**

Accept (Poster)

**Comment:**

Clearly explained, well motivated and empirically supported algorithm for training deep networks while simultaneously learning their sparse connectivity.
The approach is similar to previous work (in particular Welling et al., Bayesian Learning via Stochastic Gradient Langevin Dynamics, ICML 2011) but is novel in that it satisfies a hard constraint on the network sparsity, which could be an advantage to match neuromorphic hardware limitations.